# Research

physical chemistry/materials science/nuclear chemistry

muscovite, γ-ray irradiation, $H_2O$ radiolysis, dehydroxylation, surface hydrophilicity

**Authors for correspondence:**
Honglong Wang
e-mail: wanghonglong915@163.com
Ming Zhang
e-mail: mz10001_mzhang@sina.com

†Present address: Yinhe 596 Campus, Shuangliu, Chengdu 610200, Sichuan, People's Republic of China

This article has been edited by the Royal Society of Chemistry, including the commissioning, peer review process and editorial aspects up to the point of acceptance.

# Intensive study on structure transformation of muscovite single crystal under high-dose γ-ray irradiation and mechanism speculation

Honglong Wang[†], Yaping Sun, Jian Chu, Xu Wang and Ming Zhang[†]

Institute of Materials, China Academy of Engineering Physics, Jiangyou 621908, Sichuan, People's Republic of China

(iD) HW, 0000-0002-4811-8222

Intensive study on structure transformation of muscovite single crystal under high-dose γ-ray irradiation is essential for its use in irradiation detection and also beneficial for mechanism cognition on defect formation within a matrix of clay used in the disposal of high-level radioactive waste (HLRW). In this work, muscovite single crystal was irradiated with Co-60 γ ray in air at a dose rate of $54\,Gy\,min^{-1}$ with doses of 0–1000 kGy. Then, structure transformation and mechanism were explored by Raman spectrum, Fourier-transform infrared spectrum, X-ray diffraction, thermogravimetric analysis, CA, scanning electron microscope and atomic force microscopy. The main results show that variations in the chemical/crystalline structure are dose-dependent. Low-dose irradiation sufficiently destroyed the structure, removing Si–OH, thus declining hydrophilicity. With dose increase up to 100 kGy, CA increased from 20° to 40°. Except for hydrophilicity variation, shrink occurred in the (004) lattice plane which later recovered; the variation range at 500 kGy irradiation was 0.5% close to 0.02 Å. The main mechanisms involved were framework break and $H_2O$ radiolysis. Framework break results in Si–OH removal and $H_2O$ radiolysis results in extra OH introduction. The extra introduced OH probably results in Si–OH bond regeneration, lattice plane shrink and recovered surface hydrophilicity. The importance of framework break and $H_2O$ radiolysis on structure transformation is dose-dependence. At low doses, framework break seems more important while at high doses $H_2O$ radiolysis is important. Generally, variations in the chemical structure and surface property are nonlinear and

less at high doses. This indicates using the chemical structure or surface property variation to describe irradiation is correct at low doses but not at high doses. This finding is meaningful for realizing whether muscovite is suitable for detecting high-dose irradiation or not, and mechanism exploration is efficient for identifying the procedure for defect formation within the matrix of clay used in disposal HLRW in practice.

## 1. Introduction

Nuclear energy is a highly efficient energy being widely used in the world. In addition to being a source of power, benefiting economy and defence, two features are crucial for sustainable development of this energy. One is sufficient shielding or detecting irradiation as irradiation is accompanied by nuclear energy (e.g. nuclear weapon explosion or test, nuclear accident) and is very dangerous [1]. The other is efficient disposal of waste, especially of high-level radioactive waste (HLRW), which is toxic and radioactive [2–4].

In the field of irradiation detection, for good stability and low cost, muscovite is proposed as a detector [5–9]. Its sensibility has been explored at normal doses (less than 300 kGy) [8,10] while sensibility at higher doses is rarely studied, which demands more study. In addition to detecting low-dose irradiation, detecting high-dose irradiation is also important. This is because numerous factors display high-dose irradiation (e.g. nuclear accident, HLRW, spent fuel facility, outer space). In this case, designing new material or evaluating the existing material to ensure whether it is suitable for detecting high-dose irradiation or not is useful. Nowadays, numerous materials have been evaluated such as polymers, semiconductors (silicon), glass and calcium fluoride ($CaF_2$) [11]. However, they are probably not useful because of certain disadvantages. For instance, the polymer is easy to degrade and can be heated by irradiation [12–14], the semiconductor can easily conduct electricity [12], the composition of glass is complex, $CaF_2$ generates toxic gas under irradiation [12]. In this case, designing a material which is suitable for detecting high-dose irradiation is still challenging. Besides low cost and stability, muscovite has partial advantages such as good electric insulation, heat isolation [15] and transparency. These advantages are beneficial for storing accumulated effects and observing ion track especially for ion irradiation [5,16]. It may have potential application in high-dose irradiation detection [17]. Thus, a clear knowledge of its sensibility at high-dose irradiation is useful.

For disposal of HLRW, deep geological disposal is recommended [18]. In this project, clay is proposed as a backfill material to prevent radionuclide migration [19]. Besides retention [20,21], it would uptake water and endure various irradiations [22,23]. Irradiation could destroy the structure of the matrix [24] and impurity and lead to $H_2O$ radiolysis [25,26], thus deteriorating retention capability and mechanical properties. When the matrix structure is destroyed, retention capability and mechanical properties cannot be maintained [27]. In this case, partial radionuclides might migrate to groundwater, which is dangerous [28,29]. Naturally, clay is the ultimate medium to reduce danger to the ecosystem from HLRW except for rock for the disposal project. Its radiation resistance is crucial for ensuring the effectiveness of the disposal project. In this case, a clear evaluation of the stability and mechanism exploration for the clay under irradiation is meaningful. Numerous groups have carried out research in this field (e.g. $Cs^+$, $UO_2^+$ diffusion [30,31], $Fe^{3+}$ reduction [24,32], $H_2O$ radiolysis [26,33–35]). In reality, owing to attributes such as low cost, good fire resistance and nontoxicity, clay is a widely used material, environmental and medical science for engineering material manufacture [36–41], sewage treatment or environmental remediation [42–44], drug delivery [45], etc.

To date, the main mechanism and stability are not clearly understood for the disposal of HLRW due to the complexity of material composition and environmental conditions. Normally, clay is a composite containing numerous impurities like oxides, organics, which account for as much as 40% [46]. Additionally, $H_2O$ normally exists in the material itself or in the environment. On exposure to irradiation, matrix, impurity and $H_2O$ would generate numerous radicals. They might react with each other leading to a complex product. Thus, mechanism exploration and stability evaluation are difficult [47].

In reality, the main property of clay is that of a matrix. Having a clear understanding of stability and mechanisms for defect formation within the clay matrix is useful. The clay matrix is a phyllosilicate and defects in phyllosilicate are similar [48]. Taking these factors into consideration, using pure phyllosilicate crystal to speculate radiation damage or to explore mechanisms for defect formation within the matrix of clay used in the disposal of HLRW in practice is useful. To make the system understandable and

comparable, the sample should be pure and close to the structure of the matrix used in practice. In this condition, muscovite is probably more suitable.

Muscovite-$KAl_2(AlSi_3O_{10})(OH)_2$-single crystal is a pure phyllosilicate crystal showing a '2 : 1' layered structure (T−O−T). Two $SiO_4$ tetrahedron sheets are linked together by an $AlO_6$ octahedron sheet forming a tri-layer structure. One-fourth tetrahedral Si are substituted by Al and one-third octahedron sites show vacancies [49,50]. Vacancy shows as a dioctahedral structure in the octahedron sheet. Normally, the tri-layer structure is linked together by $K^+$ ion via weak ionic bonds [50–53]. For strong assemblage of adjacent layers, it is non-expandable. Although muscovite is not proposed to be used for the disposal of HLRW, its layered structure is similar to the matrix of clay used for this purpose. Simultaneously, the amount of impurity and $H_2O$ in this material is low, thus reducing the complexity of the material component and making the system simple, benefiting understanding. In this case, the obtained variation could be mainly ascribed to the matrix, benefiting the speculation for defect formation. Normally, $H_2O$ is difficult to be removed completely [20,34,54], and its radiolysis cannot be avoided. Additionally, the level of radiation damage is up to ray species. For clay used for the disposal of HLRW, γ-ray irradiation seems more important due to its strong penetration. For strong penetration, γ ray can even penetrate packing material and is widely used for radiation modification [55–57]. Exploring the effect of high-dose γ-ray irradiation on muscovite seems more meaningful.

Therefore, in this work, muscovite single crystal was irradiated with Co-60 γ ray in air at a dose rate of 54 Gy $min^{-1}$ with doses up to 1000 kGy. Then, the variation in the structure and intrinsic mechanism was investigated. The main objectives of this work were to (1) explore the sensibility of muscovite under high-dose irradiation to ensure whether it is suitable for detecting high-dose irradiation or not, (2) understand the mechanism for structure transformation, and (3) better understand the mechanism for defect formation within the matrix of clay used in the disposal of HLRW. The main results show muscovite to be sensitive to low-dose irradiation but not to high dose. Using this material to detect high-dose irradiation is improper. $H_2O$ radiolysis is essential especially at high doses.

# 2. Experimental section

## 2.1. Materials

Muscovite-$KAl_2(AlSi_3O_{10})(OH)_2$-single crystal film (optically transparent, light pink) was bought from the University of Cambridge, U.K. Its accurate composition has been analysed and reported as MUS2 in the literature [58,59].

## 2.2. Sample preparation and irradiation

Prior to irradiation, a film with a thickness less than 200 μm was dried at 65°C for 5 h to remove the absorbed water. Then, the film was wrapped with aluminium foil and irradiated with Co-60 γ ray in air at the Institute of Nuclear Physics and Chemistry, China Academy of Engineering Physics (Mianyang, China) at room temperature, with a dose rate of 54 Gy $min^{-1}$ with doses up to 1000 kGy. Then, the samples were stored in air at room temperature before characterization.

## 2.3. Characterization

### 2.3.1. Raman spectrum

Raman spectrum (RS) experiments were carried out on a Nicolet ALMEGA XR Instrument from 90 to 1300 $cm^{-1}$ with a spectrum resolution of 0.9 $cm^{-1}$, a 532 nm laser source and a power of 4.5 mW.

### 2.3.2. Fourier-transform infrared spectrum

Reflection-model experiments were performed on a Bruker Tensor 27 spectrometer from 400 to 4000 $cm^{-1}$ with a spectrum resolution of 4 $cm^{-1}$, 32 scans per spectrum. Transmission-mode experiments were performed on a Thermo Fisher Nicolet iS50 spectrometer from 2400 to 4000 $cm^{-1}$ with a spectrum resolution of 2 $cm^{-1}$, 32 scans per spectrum. The sample prepared for the transmission-mode experiment was cut into a square with a size of 16 mm × 18 mm, and the spectrum was normalized by mass as 10 mg.

### 2.3.3. X-ray diffraction

X-ray diffraction (XRD) experiments were carried out on a D8 Advances X-ray powder diffractometer by using Cu $k_\alpha$ irradiation ($\lambda = 0.15418$ nm) with a voltage of 40 kV and a current of 40 mA. Step size and scanning 2-theta ($2\theta$) were set to be $0.02°$ and $5-90°$, respectively, and all patterns were analysed by Jade 5 software.

### 2.3.4. Thermogravimetric analysis

Thermogravimetric analysis (TGA) experiments were carried out on a Netzsch STA 449 F3 instrument from 50 to 500°C with a heating rate of 10°C min$^{-1}$, and an argon flow of 50 ml min$^{-1}$.

### 2.3.5. Contact angle analysis

Static contact angle (CA) experiments were performed on an XG-CAM contact Angle Meter. A drop of 2 μl purified water was dropped onto the sample surface and a picture was recorded by a camera immediately [60]. Then, CA was calculated and each sample was measured five times at different locations to obtain the average datum [61,62].

### 2.3.6. Scanning electron microscope

Scanning electron microscope (SEM) measurements were carried out on a Zeiss MERLIN Compact 14184 instrument with an acceleration voltage of 8 kV. Prior to the measurement, a thin layer of gold was coated on the sample surface to increase electrical conductivity [63].

### 2.3.7. Atomic force microscopy

Atomic force microscopy (AFM) experiments were performed on an NTEGRA Prima instrument (NT-MDT Co.). Prior to the measurement, a thin layer of double-faced adhesive tape was fixed on a glass slide, then the film was fixed on the tape. The tapping mode of scanning was adopted and datum was analysed by Nova-px software.

## 3. Results and discussion

### 3.1. Chemical structure analysis

Raman and Fourier-transform infrared (FT-IR) spectra are widely used to characterize the chemical structure. Figure 1 shows Raman spectra of pristine and irradiated muscovite. All spectra show three characteristic bands at 264, 409 and 704 cm$^{-1}$ corresponding to vibrations of isosceles triangle O–H–O [49], AlO$_6$ octahedron [49] and SiO$_4$ tetrahedron [50], respectively. Other bands near 196, 638, 753, 913, 955 and 1117 cm$^{-1}$ are assigned in table 1 and are related to vibrations of the tetrahedron and octahedron framework in nature [49,50,52]. Seeing spectra in macro, it might imply great variation in the 1000 kGy-irradiated sample for weak intensity. However, that is incorrect as shown by the graph. The graph (1000 kGy-irradiated sample) is generally similar to curves of other samples. It seems to have no obvious change in the band position and shape for the sample after irradiation. Simultaneously, from table 1, it is difficult to assign major bands accurately. For instance, the band at 913 cm$^{-1}$ can be assigned to breathing vibration of the tetrahedron sheet [50], or Si–O–Al vibration [52] or libration of Al$_2$–OH [49,50]. In this case, RS was not analysed quantitatively but qualitatively.

No internal band disappeared or extra band appeared after irradiation in the Raman spectra, indicating a slight variation in the framework. This can be explained as follows. Normally, RS is sensitive to nonpolar vibration such as vibration of the C–C bond [64]. In this case, for muscovite, the spectrum mainly reflects Si–O or Al–O vibration. Irradiation is efficient at destroying the chemical bond [62,63], while for muscovite, its TO$_4$ might be stable. This is because the broken Si, Al or O atoms cannot leave from their positions because of the linkage of adjacent atoms. In this case, the broken bonds (Si–O or Al–O bonds) can even be regenerated. Finally, the species of the chemical bond within the tetrahedron sheet seems to vary slightly after irradiation. For the AlO$_6$ octahedron, OH vibration was observed, while the affiliation is uncertain. For instance, the band at 409 cm$^{-1}$ can be assigned to the Al$_2$–OH libration [49] or SiO$_4$ bending [52]. Additionally, variation for AlO$_6$

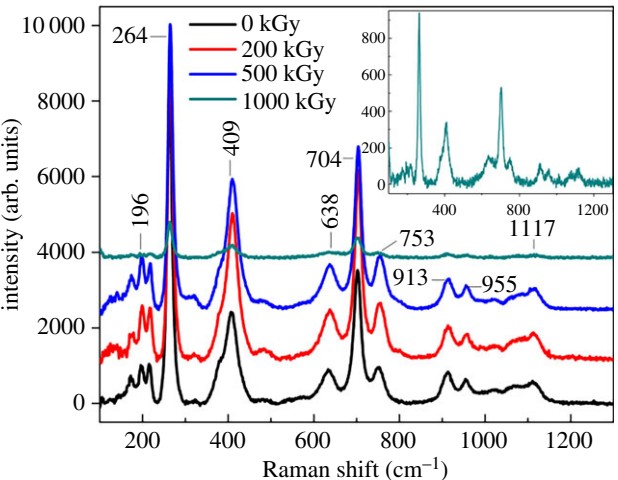

**Figure 1.** Raman spectra for muscovite under γ-ray irradiation at 0–1000 kGy.

**Table 1.** Observed Raman vibration and its muscovite assignment. Note: $O_{nb}$ = non-bridge O; $O_{br}$ = bridge O.

| band position (cm$^{-1}$) | assignment |
|---|---|
| 196 | AlO$_6$ octahedron vibration [49], Al$-$OH stretch [50,52] |
| 264 | internal vibration of isosceles triangle O$-$H$-$O [49] |
| 409 | Al$_2$$-$OH libration [49], SiO$_4$ bending [52] |
| 638,704 | internal symmetric stretching vibration of SiO$_4$, Al$-$O$_{nb}$ stretch [50], δ O$-$Al$-$O vibration [52] |
| 753 | O$_{nb}$(a)$-$Al$-$O$_{nb}$(b) bending [50,52] |
| 913 | TO$_4$ breathing vibration [50], Si$-$O$-$Al vibration [52] or libration of Al$_2$$-$OH [49,50] |
| 955 | T$-$O$_{nb}$ stretch [50] |
| 1117 | T$-$O$_{br}$ stretch [50], Si$-$O$-$Si stretch [52] |

octahedron seems less. Normally, γ-ray irradiation leads to obvious variation in the chemical structure when partial atoms are removed or there is extra species participation [26,33,62,65,66]. That means numerous Si–O or Al–O bonds destroyed or new bonds (e.g. Si–C or Si–N bonds) formed could alter RS obviously. In reality, Si–O or Al–O bonds cannot be effectively destroyed due to their intrinsic characteristic. Simultaneously, to ensure the sample's purity, coming into contact with no extra new elements during the irradiation process, the chemical bond with an extra element cannot be formed. In this case, the species of the chemical bond in the framework varied slightly after irradiation. These assumptions could probably explain the slight variation in Raman spectra.

The FT-IR spectrum is efficient in characterizing asymmetric vibration and describing OH vibration more clearly, which are informative. Figure 2 shows FT-IR spectra obtained by the reflection model for muscovite under γ-ray irradiation at 0–1000 kGy. Several bands at 685, 744, 803, 895 and 967 cm$^{-1}$ were observed mainly related to vibrations of the tetrahedron sheet (Si–O or Al–O bonds, table 2). It seems that there is no obvious change in this region. A band near 3623 cm$^{-1}$ was also observed, corresponding to the Al–OH stretch [49,58,59]. Additionally, two shoulder bands at 3695 and 3734 cm$^{-1}$ were observed corresponding to the Al–Al–OH stretch and Si–OH vibration in nature [59]. All three bands are related to O–H vibration. It seems that there is a partial change in this region. For the pristine sample, a band at 3734 cm$^{-1}$ assigned to Si–OH vibration was observed. After irradiation, this band disappeared at low dose then reappeared at a dose higher than 500 kGy. Simultaneously, the vibration of the Al–Al–OH stretch at 3695 cm$^{-1}$ was enhanced under 1000 kGy irradiation. This phenomenon is interesting and can be explained as follows. On irradiation, the Si–OH bond normally existing in silicate such as the SiO$_2$ particle [67,68] can be destroyed, showing reduced Si–OH vibration. Except for destruction, broken parts can be restored as the radiation effect is not linear with the absorbed dose [69,70]. In this case, the Si–OH band can be observed in the high-dose-irradiated sample. Except for the Si–OH band, the Al–Al–OH band varied showing variation in the octahedron (table 2).

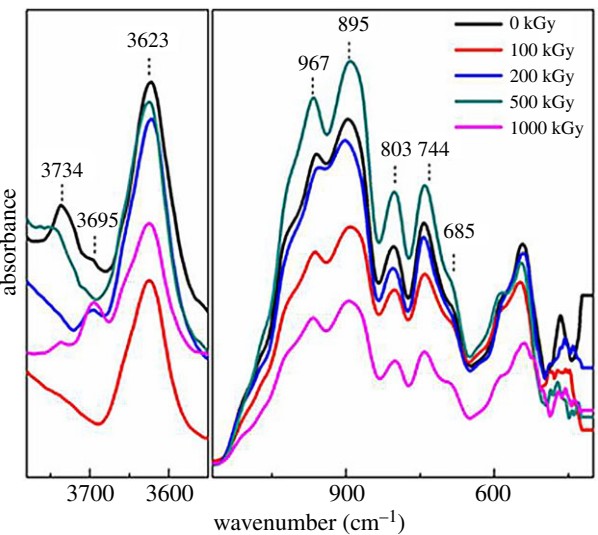

**Figure 2.** FT-IR spectra obtained by the reflection model for muscovite under $\gamma$-ray irradiation at 0–1000 kGy.

**Table 2.** Observed FT-IR vibration and its muscovite assignment. Note: $O_{nb}$ = non-bridge O.

| band position (cm$^{-1}$) | assignment |
|---|---|
| 685 | SiO$_4$ vibration as $\delta$ Si−O−Al, $\delta$ Si−O−Si [52], or Al−O$_{nb}$ [50,58] |
| 744 | Al−O−Si vibration and other [58] |
| 803 | Al−O motion or stretch, Al−O−Al bending or stretch [50,52,58] |
| 895 | Al−OH bending [58], Al−O−Al libration [52] |
| 967 | SiO$_4$ vibration [58], Si−O−Si stretch [52,58] |
| 3623 | Al−OH stretch [58,59] |
| 3695 | Al−Al−OH stretch [59] |
| 3734 | Si−OH vibration [59] |

From figure 2, partial change in OH vibration can be seen, which needs quantitative analysis. Nevertheless, partial difficulties exist. Firstly, the baseline at 3500–3800 cm$^{-1}$ is not straight. Secondly, the vibration for the tetrahedron is complex. We cannot assign accurate vibration. For instance, the band at 967 cm$^{-1}$ can be categorized as SiO$_4$ vibration [58], the Si−O−Si stretch [52,58]. Additionally, this band is a complex overlay. It is difficult to split. Finally, it is difficult to assign an internal standard band. Thus, it is difficult to analyse quantitatively.

To describe OH vibration clearly, films were measured on another FT-IR spectrometer by transmission-mode with a square of size of 16 mm × 18 mm and normalized by mass as 10 mg.

Figure 3 shows FT-IR spectra obtained by transmission-mode for muscovite from 1800 to 4000 cm$^{-1}$ with normalization by mass as 10 mg. From figure 3a, it seems that there is no obvious change in band position and shape and only a band near 3627 cm$^{-1}$ was observed corresponding to Al−OH vibration in the octahedron [58]. This is inconsistent with the FT-IR spectrum obtained by the reflection model as Si−OH vibration (3734 cm$^{-1}$) observed in that spectrum (figure 2). This is probably because of the difference between the two methods used and the sample structure. For muscovite, its OH vector in the z-direction is strong (the included angle between the z-direction and OH is less than 30°) [71], and Si−OH vibration is weaker compared to Al−OH (figure 2). For the FT-IR transmission-mode experiment, the propagation direction of a photon is vertical to the sample surface. In this case, the signal for OH vibration is weak. As Si−OH vibration is weaker compared to Al−OH, in this case, the signal is slight. Nevertheless, for the FT-IR reflection-model experiment (figure 2), the propagation direction of a photon is not vertical to the sample surface, but probably has an angle of 45° incline. In this case, the signal for OH vibration can be strong and Si−OH vibration can be observed. These assumptions could probably explain the conflict. To describe OH amount clearly, the band area near 3627 cm$^{-1}$ was integrated and is shown

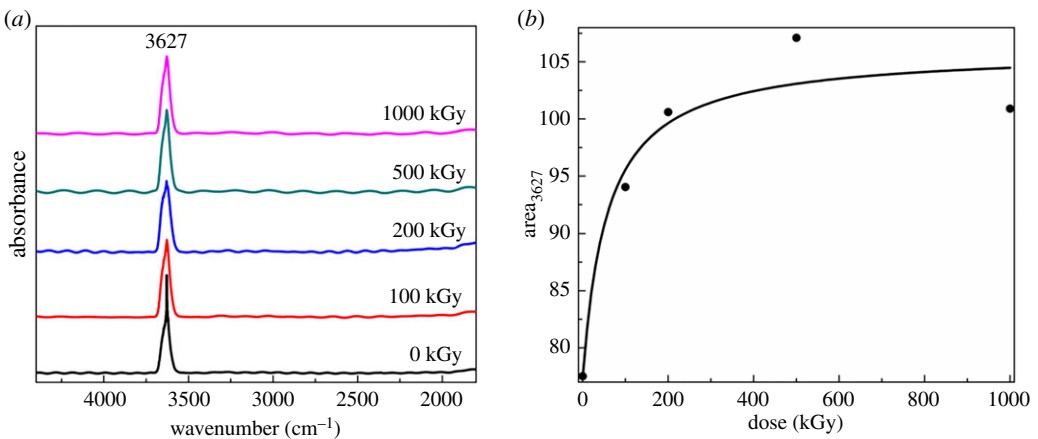

**Figure 3.** FT-IR spectra obtained by transmission-mode for muscovite from 1800 to 4000 cm$^{-1}$ with normalization by mass as 10 mg.

in figure 3*b*. Generally, the band area increased with the increase in dose. For the 1000 kGy-irradiated sample, the band area was larger than the pristine sample by nearly 20%, showing that numerous Al–OH were introduced, meaning dehydroxylation was not the dominant reaction during the irradiation process. As the sample did not contact other species except air during the irradiation process, the extra OH pull-in was probably due to $H_2O$ radiolysis as partial $H_2O$ was lying on the surface, interlayer or edges [34]. In reality, Al–OH bonds mainly exist in the octahedron sheet. The extra introduced Al–OH bonds are probably mainly derived from the breakage of links between tetrahedron and octahedron sheets (Al–O–Si or Al–O–Al bonds). In this case, partial Si–OH bonds can be generated coupled with Al–OH bond generation. This assumption could probably support the Si–OH band reappearance in the high-dose-irradiated sample. The extra introduced chemical bond might alter the crystalline structure, and $H_2O$ radiolysis rate could decline.

Normally, quantitatively analysing FT-IR spectrum should assign an internal standard band. Using mass to quantitatively analyse the FT-IR spectrum is unusual. In reality, we have adopted that strategy. Nevertheless, the band at a low wavenumber is easy to saturate and the vibration for OH is weak. In this case, when the signal for OH is visual, the signal at low wavenumber is saturated when the signal at the low wavenumber is proper, and the signal for OH is weak. It is difficult to use an internal standard band to quantitatively analyse the OH amount in the FT-IR spectrum obtained by the transmission-mode experiment. Thus, the spectrum was normalized by mass. We think this strategy is also effective.

Generally, Raman and FT-IR analysis indicate irradiation had no intense effect on the matrix while having an obvious effect on the microstructure. Upon irradiation, the Si–OH bond was destroyed at a low dose then regenerated at a high dose. Simultaneously, an extra Al–OH bond was introduced, probably ascribed to $H_2O$ radiolysis. The extra Al–OH group's introduction indicates dehydroxylation was not dominant.

## 3.2. Crystalline structure analysis

The extra introduced OH might alter the crystalline structure, probably resulting in crystal shrink or expansion. Correspondingly, the variation in the crystalline structure could reflect the variation in the chemical structure, benefiting mechanism exploration. Figure 4*a* shows XRD patterns for muscovite under γ-ray irradiation at 0–1000 kGy. The main lattice planes were assigned by Jade 5 software according to standard PDF cards. For the pristine sample, there were five main lattice planes assigned as (002), (004), (006), (008) and (224) with the corresponding 2θs as 8.88°, 17.76°, 26.81°, 36.00° and 45.43°, respectively. For irradiated samples, patterns were similar to the pristine sample. For all samples, several peaks with 2θs as 55.19°, 64.43° and 76.28° were observed, probably due to lattice plane repetition as the method used was powder diffraction while sampling in film. Generally, XRD patterns did not show an obvious change in peak shape and position for the sample after irradiation, implying no obvious variation in species. If species were altered, patterns would vary obviously as different materials or phases have distinct lattice plane parameters. A similar pattern indicates no serious decomposition or phase transformation has occurred during the irradiation process. Although no serious transformation occurred, partial variation in micro probably occurred.

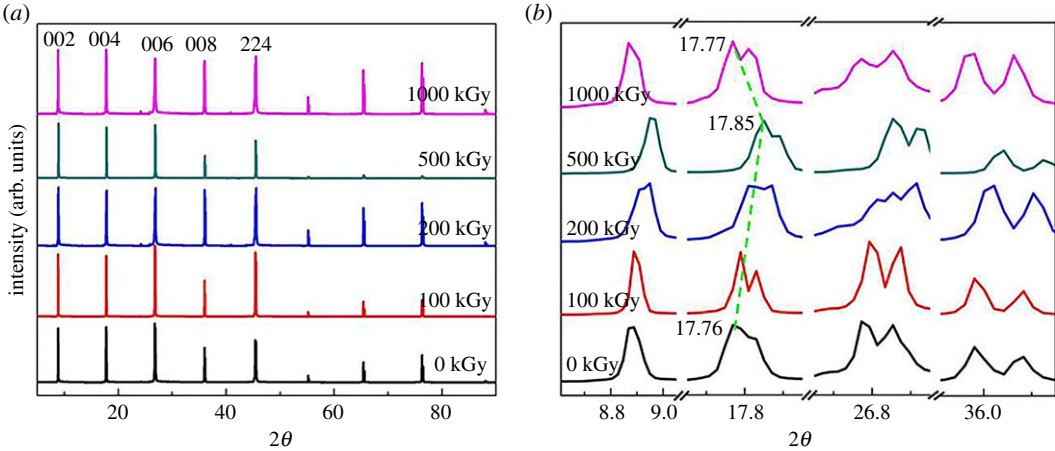

**Figure 4.** XRD patterns for muscovite under γ-ray irradiation at 0 − 1000 kGy.

Figure 4*b* shows refined XRD patterns for (002), (004), (006) and (008) lattice planes. All patterns are similar while generally shifting to a higher angle at low dose than recovered at a dose higher than 500 kGy. Taking the (004) lattice plane as a representative, pristine, 500 kGy-irradiated and 1000 kGy-irradiated samples had $2\theta$s as 17.76°, 17.85° and 17.77°, respectively. According to Bragg's formula ($n\lambda = 2d\sin\theta$), for the refined lattice plane and measurement condition, $n$ and $\lambda$ are constant. In this case, increased $\theta$ means reduced $d$ [72]. In other words, low-dose irradiation declines the interlayer space, resulting in shrinking.

To quantitatively describe this variation, interlayer space $d$ for the (004) lattice plane was investigated. For the aforementioned $2\theta$s, the ratio of the interlayer space for 500 kGy-irradiated and pristine sample ($d_{500}/d_0$) is 99.4% ($d_{500}/d_0 = \sin\theta_0/\sin\theta_{500} = \sin8.88°(17.76/2)/\sin8.93°(17.85/2) = 0.9944 \times 100\% = 99.44\%$), meaning 0.5% ($100-99.44\% \approx 0.5\%$) shrink under 500 kGy irradiation. The ratio of the interlayer space for 1000 kGy-irradiated and pristine samples ($d_{1000}/d_0$) is 99.9% ($d_{1000}/d_0 = \sin\theta_0/\sin\theta_{1000} = \sin8.88°(17.76/2)/\sin8.89°(17.77/2) = 0.9989 \times 100\% = 99.89\%$), meaning interlayer space $d$ was recovered by extra irradiation. To understand variation more accurately, interlayer space $d$ for this lattice plane was calculated by Jade 5 software on the aforementioned $2\theta$s and $\lambda = 0.15418$ nm. $d_0$, $d_{500}$ and $d_{1000}$ were $4.985 \pm 0.002$, $4.962 \pm 0.002$ and $4.981 \pm 0.004$ Å, respectively, and $d_{500}/d_0 = 4.962/4.985 \approx 0.9954 \approx 99.54\%$, $d_{1000}/d_0 = 4.981/4.985 \approx 0.9991 \approx 99.91\%$. The result is close to that compared by $\sin\theta$. Generally, under 500 kGy irradiation, the (004) lattice plane shrinks nearly by 0.5%, closing to 0.02 Å ($4.985 (\pm 0.002) - 4.962 (\pm 0.002) = 0.023 (\pm 0.002)$ Å).

Seeing the value, the declined range is small while the phenomenon is interesting and the intrinsic mechanism seems to be important. The decline of the interlayer space means shrinking of the lattice plane. In most cases, the variation is within the octahedron sheet as this layer has numerous vacancies [49,50,52] and is unstable compared to the tetrahedron sheet. Owing to vacancies, partial OH is linked to the non-bridge O atom forming hydrogen bonds, shown as isosceles triangle O−H−O [49,73]. As a result, the OH vector is not parallel to the *z*-direction. The formation of hydrogen bond strengthens the interlayer force within tetrahedron and octahedron sheets [74], shrinking the interlayer space. That means extra hydrogen bond formation might result in shrinking. Normally, the chemical bond break results in shrinking of the crystalline unit. On the contrary, extra chemical bond introduction expands the lattice plane [75]. From the aforementioned analysis upon 500 kGy irradiation, interlayer space $d$ of the (004) lattice plane declined, showing shrinking. This is interesting. This is probably mainly due to framework break and hydrogen bond formation. The broken bonds are probably the linkage between tetrahedron and octahedron sheets as Al−O−Si or Al−O−Al bonds as the break of the Al−OH bond would destroy hydrogen bond resulting in expansion for muscovite [58]. The break of the chemical bond in the tetrahedron sheet would not alter the interlayer space as the interlayer space mainly lies on the T−O−T structure in the *z*-direction. Naturally, it is the scale of the Si(Al)−O−Al−O−Si(Al) bond length in the *z*-direction. Except for chemical bond break, an extra Al−OH bond was introduced (figure 3). These might form a hydrogen bond, benefiting shrinking. These assumptions could probably explain lattice plane shrink at low doses. At high doses, the extra introduced OH might be destroyed and reject each other. In this case, the interlayer space can recover. These assumptions could probably explain recovery of interlayer space $d$ for the 1000 kGy-irradiated sample.

The variation range (near 0.5%, close to 0.02 Å) may be small compared to variation induced by ionizing irradiation such as $Au^{3+}$, $Pb^{2+}$ or $He^{2+}$ ions as they are huge in volume and charge, showing a high linear energy transfer (LET) effect easily inducing the displacement of atoms in a lattice [70,76]. Finally, they easily induce phase transformation, amorphization or decomposition. However, the variation range (0.5%) may be obvious for γ-ray irradiation as LET effect for γ ray is weak. For low LET effect, it is difficult to deposit huge energy in lattice. Thus, the temperature elevation in lattice and atom vibration enhanced by temperature elevation can be ignored. In this case, γ-ray irradiation cannot result in the displacement of atoms efficiently, only via random ionization or motivation can the chemical bond be broken to affect the crystalline structure. Thus, variation in lattice induced by γ-ray irradiation can be tiny. Normally, for solid material, a tiny variation in lattice could alter the macroscopic property obviously. Thus, the effect of variation range near 0.5% cannot be ignored. In reality, the refined pattern for the (006) lattice plane with $2\theta$ near 26.81° varied obviously, especially for the 200 kGy-irradiated sample. The pattern for this sample also showed split (figure 4b).

Generally, XRD experiments show that the (004) lattice plane shrunk near 0.5% upon 500 kGy irradiation. The main reasons probably derive from framework break and hydrogen bond formation. Lattice plane shrink also indicates dehydroxylation was not dominant, confirming the FT-IR results.

## 3.3. $H_2O$ amount analysis

The extra introduced Al−OH bond certified by the FT-IR spectrum was probably due to $H_2O$ radiolysis. The shrink of the (004) lattice was probably due to extra hydrogen bond formation. In this case, the $H_2O$ amount should decline. It is essential to certify this process. In early reports [58,77–85], for muscovite, thermal-induced dehydroxylation would not happen at a temperature lower than 500°C. Under this assumption, mass variation at a temperature lower than 500°C could be ascribed to volatilization of $H_2O$ that originally existed. Thus, TGA could characterize $H_2O$ amount variation.

Figure 5 shows TGA curves of pristine, 500 kGy-irradiated and 1000 kGy-irradiated muscovite, respectively. All curves show a similar trend. With temperature increases up to 500°C, the mass reduced slightly. For instance, the mass of pristine, 500 kGy-irradiated and 1000 kGy-irradiated samples decreased to 95.7%, 97.8% and 96.1%, respectively. Simultaneously, curves do not show a sharp decline, indicating no intense volatilization of organics or matrix decomposition occurred during measurement. Assuming the sample is pure without impurity except for $H_2O$, the slight decline could be ascribed to $H_2O$ volatilization as $H_2O$ generally exists in clay [34,86] and can be evaporated at 50−500°C [34,87,88].

Simultaneously, we hypothesize that the volatilization of $H_2O$ is linear with its content and $H_2O$ amount within the sample before irradiation. In this case, variation in mass reduction could be related to the irradiation process. The pristine sample has the largest mass reduction at 4.3% (100 − 95.7% = 4.3%). Five hundred and 1000 kGy-irradiated samples have mass reductions of 2.2 (100 − 97.8% = 2.2%) and 3.9% (100 − 96.1% = 3.9%), respectively. In other words, the $H_2O$ amount in pristine, 500 kGy-irradiated and 1000 kGy-irradiated samples can be considered to be 4.3%, 2.2% and 3.9%, respectively, showing a declined $H_2O$ amount in the irradiated sample.

After irradiation, the $H_2O$ amount declined, which is interesting. To our knowledge, γ-ray irradiation is a cold irradiation model compared to ionizing irradiation like electron-beam irradiation [89], and cannot elevate sample temperature efficiently. In this case, the volatilization of $H_2O$ induced by temperature elevation related to irradiation during the irradiation process can be ignored. Because of this, the reason for the decline in amount of $H_2O$ could be ascribed to its radiolysis as $H_2O$ is easy to radiolyse [25,90] and radiolysis products-H· and HO· radicals are reaction-active, reacting easily with the framework introducing groups like Al−OH. The extra introduced OH could enhance the OH signal in the FT-IR spectrum and might form hydrogen bond resulting in lattice plane shrink. In this case, TGA analysis further confirmed FT-IR and XRD results.

The decline of $H_2O$ amount in the irradiated sample also indicates dehydroxylation was not the dominant reaction during the irradiation process as that reaction would raise the $H_2O$ amount. If that procedure was dominant, numerous Al−OH bonds would be destroyed, hydrogen bond (triangle O− H−O) [49] would be destroyed seriously, probably resulting in serious expansion [58]. This would be inconsistent with XRD analysis. The declined $H_2O$ amount observed indicates $H_2O$ radiolysis is essential. Although dehydroxylation was not dominant, we cannot be sure whether this procedure occurred or not as the dynamic procedure is difficult to observe *in situ*.

It seems the $H_2O$ amount in the 500 kGy-irradiated sample is lower than that in the 1000 kGy-irradiated sample and the $H_2O$ amount in the 1000 kGy-irradiated sample is close to that in the

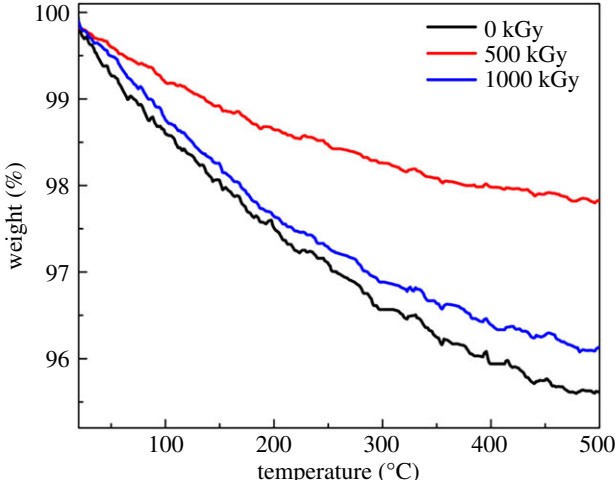

**Figure 5.** TGA curves for muscovite under γ-ray irradiation at different doses.

pristine sample. These results can support FT-IR analysis commendably as the 500 kGy-irradiated sample has the most intense signal for OH vibration (figure 3). Simultaneously, if we assume lattice plane shrink is completely ascribed to extra OH introduction, in this case the 500 kGy-irradiated sample has the most intense shrink in the lattice plane. In other words, it has the largest amount of OH introduction. The extra introduced OH is due to $H_2O$ radiolysis. Thus, the 500 kGy-irradiated sample should have lower $H_2O$ amount compared to the pristine sample. TGA analysis certified this assumption. That means TGA results supported XRD analysis indirectly. The reason for the 500 kGy-irradiated sample having lower $H_2O$ amount compared to pristine and 1000 kGy-irradiated samples can be explained as follows. Upon irradiation, two procedures occurred synchronously. One is chemical bond break such as the break of the Al−OH bond shown as dehydroxylation. The other is the extra group's introduction related to $H_2O$ radiolysis. The former procedure would raise the $H_2O$ amount, the latter would reduce the $H_2O$ amount. Finally, the mechanism for $H_2O$ amount variation is complex. At low dose, framework break is dominant and $H_2O$ radiolysis can be promoted by dose increase. At high dose, extra irradiation probably results in more break of Al−OH, and $H_2O$ can be regenerated. In this case, the 500 kGy-irradiated sample might have lower $H_2O$ amount compared to pristine and 1000 kGy-irradiated samples and the $H_2O$ amount in the 1000 kG-irradiated sample can be close to that in the pristine sample. This phenomenon is normally observed, as the radiation effect is not linear with the absorbed dose such as radiation cross-linking of the ultra-high molecule weight polyethylene sheet [65]. After a threshold, the degree of cross-linking would be constant or smaller.

Normally, FT-IR and TGA analyses indicate $H_2O$ radiolysis, probably implying extra OH introduction in the framework. In most cases, the introduction of an extra chemical bond would enlarge the volume parameter such as interlayer space $d$ as expansion occurs more easily in the $z$-direction than $x$- and $y$-directions for mica [60,91]. As seen from XRD results (figure 4), no expansion was observed. This seems contrary, while it can be ascribed to the structure. As one-third of octahedron sites are vacancies and the OH vector is not parallel to the $z$-direction but forms hydrogen bonds (triangle O−H−O) [49], its structure does not heap up compactly. Numerous spaces are idle. In this case, its lattice plane is difficult to expand intrinsically. Under OH introduction, the lattice plane would not expand efficiently. Simultaneously, chemical bond break would shrink the tetrahedron or octahedron unit, minimizing volume parameters like interlayer space $d$. Additionally, partially introduced OH might form hydrogen bonds. Finally, lattice plane shrink was observed. These assumptions could probably explain the conflict.

Generally, TGA measurements show a declined $H_2O$ amount in irradiated samples, indicating dehydroxylation was not the dominant reaction during the irradiation process. The main reason was probably ascribed to $H_2O$ radiolysis. In this case, the TGA result can certify extra Al−OH bond introduction and lattice plane shrink indirectly.

## 3.4. Surface hydrophilicity and morphology

Chemical structure variation would alter surface properties such as wettability [92]. Static CA can efficiently characterize surface hydrophilicity [62,63]. Thus, CA experiments were performed. Figure 6*a*

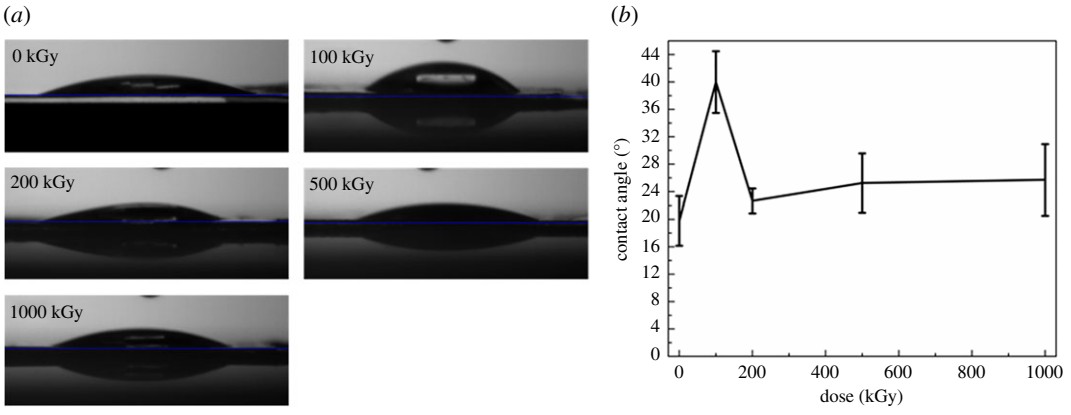

**Figure 6.** (a) Optical images of water droplets on sample surface and (b) static CAs for muscovite under γ-ray irradiation at different doses.

shows optical images of water droplets on the sample surface. For the pristine sample, the droplet of water almost spreads out completely, showing excellent hydrophilicity. For irradiated samples, the sprawl of droplet is similar to the pristine sample except for the 100 kGy-irradiated sample, also showing nice hydrophilicity. It seems that hydrophilicity declined at a lower dose than recovered by extra irradiation.

To quantitatively describe variation within hydrophilicity, CA was calculated and shown in figure 6b. The pristine sample has the smallest CA by approximately 20°. Irradiated samples have varied CAs. For instance, CA increased to approximately 40° with dose increases to 100 kGy then decreased to approximately 23° with dose increases to 200 kGy. Then, CA seems to be constant within 23–26° with dose continuously increasing to 1000 kGy. This indicates that low-dose irradiation sufficiently declined hydrophilicity. In other words, the effect of irradiation on surface hydrophilicity is dose dependent. This can be explained as follows. According to FT-IR analysis, extra OH was introduced, promoting hydrophilicity. Except for OH introduction, surface Si–OH was removed, declining hydrophilicity. During the irradiation process, the aforementioned procedures occurred synchronously and had offset effects on the obtained CA. Finally, the mechanism for CA variation is complex. In reality, Si–O, Si–OH and Al–OH bonds are hydrophilic [93,94]. In this case, the pristine sample has good hydrophilicity. Normally, surface hydrophilicity mainly relies on the surface structure which mainly contains the tetrahedron sheet and interlayers ions (e.g. $K^+$). In this case, variation in in-plane probably does not affect hydrophilicity. During the irradiation process, surface destruction cannot be avoided [25,33,95] and the Si–OH bond was removed (figure 2b). In this case, surface hydrophilicity declined at low doses. With dose increases, $H_2O$ radiolysis became serious. Partial H· or HO· radicals might react with the tetrahedron sheet, regenerating Si–OH or Al–OH bonds, promoting hydrophilicity. Additionally, the extra introduced OH or destruction might increase micro-roughness, probably enlarging the specific surface area, strengthening the interfacial force between the tetrahedron sheet and $H_2O$, promoting hydrophilicity. In this case, wettability can be recovered by extra irradiation and a spinodal was observed in CA. These assumptions could explain CA decrease with dose increase from 100 to 200 kGy.

With dose continuously increasing from 200 to 1000 kGy, the extra irradiation seems to have no effect on CA. The result is expected and consistent with the FT-IR spectrum as the increased OH amount in this region is close. Simultaneously, the Si–OH bond was regenerated slightly (figure 2b). Normally, an increase in the OH amount could promote hydrophilicity. Nevertheless, the material is hydrophilic, even with a high increase in the OH amount, CA would not decrease obviously. This phenomenon is normal. For instance, radiation grafting polyacrylic acid to increase surface hydrophilicity of a polymer or CNTs, after a threshold, the CA would increase or be constant with increase of degree of grafting. Additionally, the destruction within the tetrahedron sheet can be serious. In this case, CA seems to be constant or have slight variation.

Apart for the chemical structure, macro morphology also has a great effect on the obtained CA. For the hydrophilic material according to the Wenzel model, the larger the roughness, the smaller the CA. In this case, if there were grooves or cracks on the sample surface, CA would decrease sufficiently. Its effect might exceed the effect induced by irradiation. In this case, all aforementioned explanations for chemical structure variation would be improper. It is essential to eliminate the artificial factor. In figure 6b, CA for

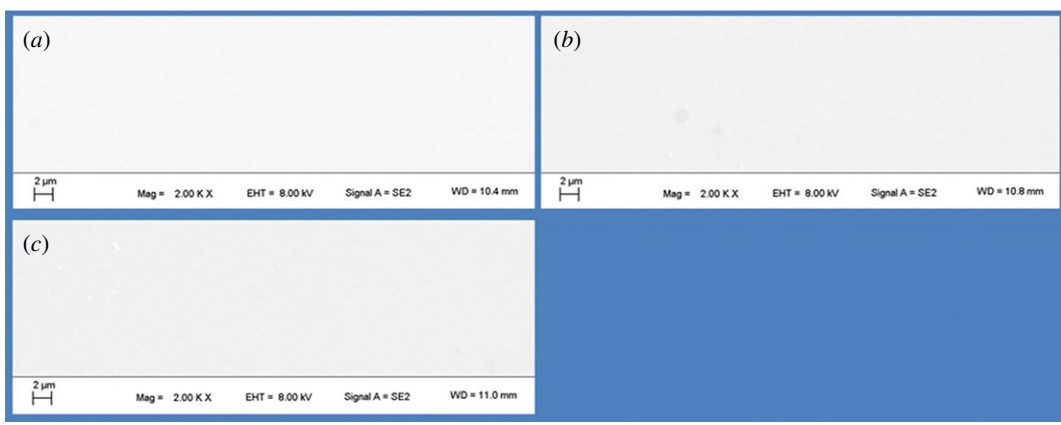

**Figure 7.** SEM images for the surface of (*a*) pristine, (*b*) 200 kGy-irradiated and (*c*) 1000 kGy-irradiated muscovite.

the 200 kGy-irradiated sample is smaller compared to the 100 kGy-irradiated sample and a spinodal exists at 100 kGy. Whether the decrease of CA for the 200 kGy-irradiated sample is due to the artificial factor or not needs confirmation. To make a comparison, pristine and 1000 kGy-irradiated samples were also observed.

Figure 7 shows SEM images for the surface of (a) pristine, (b) 200 kGy-irradiated and (c) 1000 kGy-irradiated samples, respectively. All samples display no obvious groves or cracks, showing a smooth surface. That means films prepared for CA experiments were smooth. Thus, we can eliminate the artificial factor on the obtained CA. In other words, the CA variation was not caused by sample preparation but reflected variation in the microscopic structure. That means the reason for decreased CA for the 200 kGy-irradiated sample could be ascribed to chemical structure variation. In this condition, explanations for CA variation on microstructure are appropriate. Additionally, the smooth surface also indicates $\gamma$-ray irradiation had no obvious effect on the macroscopic morphology of muscovite. This phenomenon is expected. In reality, $\gamma$ ray is a high energy photon which almost has no mass compared to the framework atom. In this case, it is difficult to induce displacement of atom directly just by random ionization or motivation. As the density of the photon is incompact and crash between the photon and framework atom is random, the LET effect is slight. In this case, it is difficult to deposit huge energy in the microcell. Thus, it is difficult to induce obvious variation in macro morphology in theory. Even under $^{197}$Au (11.4 MeV n$^{-1}$) ion irradiation for several minutes, which has an intense LET effect, variation in the z-direction is several nanometre for muscovite [7]. For $\gamma$-ray irradiation, for a weak LET effect, the variation would be smaller. In this condition, the variation could be in Å scale. This tiny scale probably exceeds resolution of SEM or AFM technology.

To observe morphology more clearly, AFM experiments were performed. Figure 8 shows AFM images for the surface of pristine and irradiated samples. Generally, variation in the z-direction is tiny as in the nanometre scale, implying smooth. To have a clear realization of the difference in the z-direction, an area is chosen randomly as shown in green colour in the image (figure 8). Then, the height in the z-direction after cropping versus distance along the green line is displayed and shown in figure 9. In figure 9a, the variation range after cropping for the pristine sample is close to $\pm 2$ nm except for the partial section. In figure 9b–d, variation ranges after cropping for 100 kGy-irradiated, 200 kGy-irradiated and 500 kGy-irradiated samples can be considered as $\pm 0.7$, $\pm 0.2$ and $\pm 0.6$ nm, respectively. In figure 9e, variation range after cropping for the 1000 kG-irradiated sample is close to $\pm 1.5$ nm. For the pristine sample (figure 9a), the partial section shows large variation range. This is probably due to an existing defect as the sample is natural and in a sheet form. In this case, the surface might show bumps or depressions induced by mining. Additionally, the film is peeled randomly. If the sample contains defects internally, partial sections can be broken during peeling. In this case, partial bumps or depressions can be observed. Except for that region, the sample is generally smooth with a variation range close to $\pm 2$ nm in the z-direction in the chosen part. For 100 kGy-irradiated, 200 kGy-irradiated and 500 kGy-irradiated samples, the variation ranges are close to $\pm 0.7$, $\pm 0.2$ and $\pm 0.6$ nm. They are very tiny, showing smooth. In reality, the size of the T–O–T structure in the z-direction is close to 1.2–1.5 nm, the chemical bond length for the K–O ionic bond or the Si–O/Al–O bond is close to 0.2 nm. Seeing the value especially for the 200 kGy-irradiated sample, the variation range is close to $\pm 0.2$ nm, which is close to the length of the chemical bond as the K–O ionic bond or Si–O bond. Normally, the distribution of the interlayer ion (e.g. K$^+$ ion) is

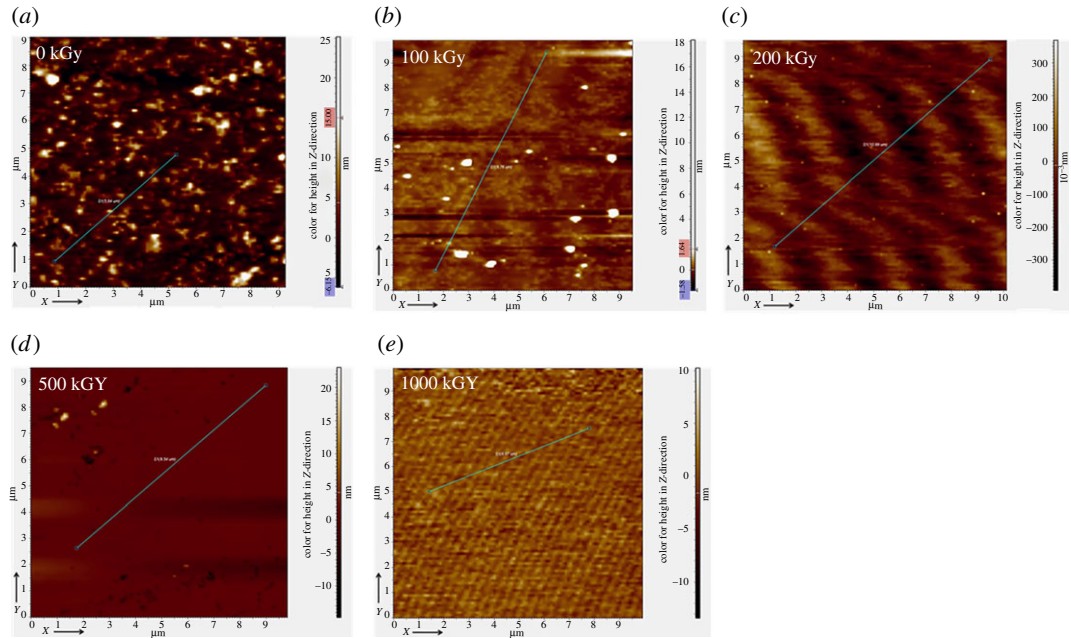

**Figure 8.** AFM images for the surface of muscovite under γ-ray irradiation at 0 – 1000 kGy.

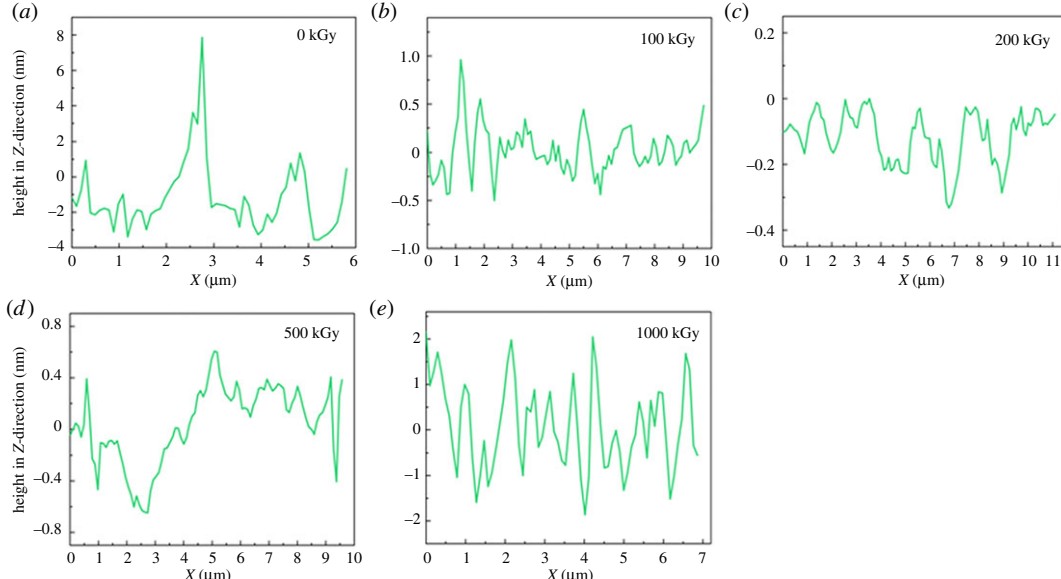

**Figure 9.** Height in the z-direction after cropping versus distance along the chosen area for different samples.

random; in this case, the variation in the z-direction at the atom level may be close to 0.2 nm. In this work, the variation range is close to this scale. That means the surface is very smooth.

Generally, for all samples, the variation range in the z-direction after cropping is less than ±2 nm. The range is tiny. Normally, the variation range induced by the artificial factor is large, which can be in μm scale or larger. Trying to induce variation in the nanometre scale by artificial technology is difficult. In reality, this tiny variation needs special instruments or technology, which should be processed precisely such as by laser etching. For samples peeled manually, this tiny variation can be endured. Generally, this tiny variation indicates the sample is smooth, which further confirms SEM results.

Although the sample is smooth, the variation range for the pristine sample seems larger than other samples. That means it seems rougher than other samples as roughness can be described by variation range in the z-direction. This can be ascribed to sample preparation or difference. For irradiated samples, they seem smoother compared to the pristine sample especially for the 200 kGy-irradiated sample. That cannot be ascribed to irradiation as it is difficult to prepare the sample manually with a constant surface structure. Simultaneously, we cannot assure that the defect in the sheet is uniform. Additionally, variation

induced by irradiation in the $z$-direction would be very tiny as seen from the aforementioned explanations (SEM part). Its variation may be covered by sample preparation or difference. In this case, apparent roughness difference cannot be ascribed to the irradiation process. If we consider the pristine sample has the largest roughness and all samples have similar chemical surface structure, in this case, irradiated samples should have larger CAs especially for the 200 kGy-irradiated sample as this sample seems to have the smallest roughness. Nevertheless, the obtained CA is contrary to this expectation. They have close CAs (figure 6). That means great variation existed in the chemical structure. In other words, CA variation was not aroused by surface morphology difference but by chemical structure variation, which means the aforementioned analysis on the chemical structure is probably suitable.

Generally, from SEM and AFM analysis, it can be concluded that the CA variation was attributed to the intrinsic structure difference and not to the artificial factor. In reality, numerous procedures could affect surface hydrophilicity such as framework break, Si–OH removal or regeneration and roughness variation. They might show offset effects on CA variation. Finally, CA did not vary linearly versus the absorbed dose.

Generally, low-dose irradiation sufficiently declined hydrophilicity while extra irradiation recovered. Irradiation had almost no effect on surface morphology.

## 3.5. Mechanism illustration

From the aforementioned analysis, it seems that variations in the chemical/crystalline structure and surface hydrophilicity are dose-dependent, and muscovite is sensitive to low-dose irradiation and not to high dose. The main reasons were explained generally. To have a clear cognition, the main mechanisms will be illustrated below. It seems that the main mechanisms involve framework breaks and $H_2O$ radiolysis. Upon irradiation, breaks occurred in chemical bonds in the $TO_4$ tetrahedron sheet (e.g. Si–OH) and the linkage between tetrahedron and octahedron sheets. Simultaneously, partial $H_2O$ occurred in radiolysis, generating H· or HO· radicals. These radicals are reaction-active, reacting with the framework (e.g. the linkage between tetrahedron and octahedron sheets or the broken Si–O bonds), leading to extra OH introduction. During the irradiation process, these procedures occurred synchronously. At low doses, destruction seems predominant. In this case, the Si–OH bond was removed and surface hydrophilicity declined. At high doses, $H_2O$ radiolysis seems predominant. In this case, the $H_2O$ amount declined, an extra Al–OH bond was introduced, Si–OH bond was regenerated, and surface hydrophilicity recovered. Additionally, partial OH might form hydrogen bonds, resulting in lattice plane shrink. To describe this procedure more clearly, several equations will be used as given below and a scheme will be displayed in figure 10. Where ≡Si(Al)–O–Al–O–Si(Al)≡ represents the T–O–T structure in the $z$-direction as interlayer space $d$ mainly reflects the scale of the Si(Al)–O–Al–O–Si(Al) bond in the $z$-direction, ≡Si(Al)–O–Si(Al)≡ represents the $TO_4$ tetrahedron sheet, ≡Si–OH represents the surface Si–OH structure and ≡Si(Al)–O–Al–OH represents the Al–OH bond in the octahedron sheet. The reason for the Al element written in tetrahedron is due to the one-fourth tetrahedral Si substituted by Al. In reality, we cannot be sure whether breaks occurred in Si–O–Si or Si–O–Al bonds in the tetrahedron sheet.

Equations (3.1)–(3.6) are reactions probably originally induced by γ-ray irradiation.

$$\equiv \text{Si–OH} \rightarrow \equiv \text{Si} \cdot + \text{HO} \cdot \tag{3.1}$$

$$\equiv \text{Si(Al)–O–Si(Al)} \equiv \rightarrow \equiv \text{Si(Al)} \cdot + \cdot \text{O–Si(Al)} \equiv \tag{3.2}$$

$$\equiv \text{Si(Al)–O–Al–O–Si(Al)} \equiv \rightarrow \equiv \text{Si(Al)} \cdot + \cdot \text{O–Al–O–Si(Al)} \equiv \tag{3.3}$$

$$\equiv \text{Si(Al)–O–Al–O–Si(Al)} \equiv \rightarrow \equiv \text{Si(Al)–O} \cdot + \cdot \text{Al–O–Si(Al)} \equiv \tag{3.4}$$

$$\equiv \text{Si(Al)–O–Al–OH} \rightarrow \equiv \text{Si(Al)–O–Al} \cdot + \text{HO} \cdot \tag{3.5}$$

and

$$H_2O \rightarrow H \cdot + \text{HO} \cdot \tag{3.6}$$

Equations (3.7)–(3.10) are reactions probably between radiolysis products.

$$\equiv \text{Si(Al)} \cdot + \text{HO} \cdot \rightarrow \equiv \text{Si(Al)} - \text{OH} \tag{3.7}$$

$$\equiv \text{Si(Al)–O} \cdot + \text{H} \cdot \rightarrow \equiv \text{Si(Al)–OH} \tag{3.8}$$

$$\text{H} \cdot + \cdot \text{O–Al–O–Si(Al)} \equiv \rightarrow \text{HO–Al–O–Si(Al)} \equiv \tag{3.9}$$

and

$$\text{HO} \cdot + \cdot \text{Al–O–Si(Al)} \equiv \rightarrow \text{HO–Al–O–Si(Al)} \equiv \tag{3.10}$$

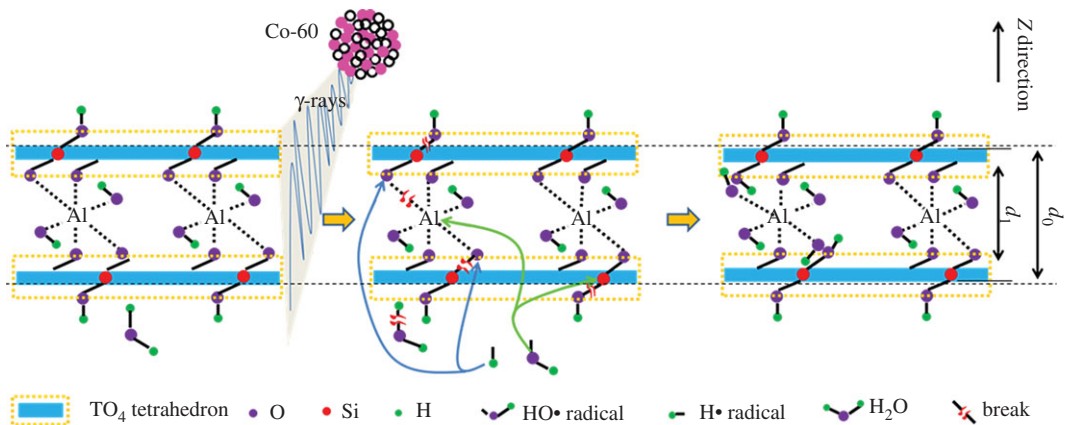

**Figure 10.** A scheme for structure transformation within the muscovite layered structure in the z-direction under γ-ray irradiation; $d_0$, interlayer space for the pristine sample; $d_1$, interlayer space for the sample after irradiation.

Generally, equations (3.1) and (3.2) describe Si−OH bond removal and tetrahedron destruction, illustrating declined hydrophilicity at low doses. Equations (3.3) and (3.4) describe the linkage break between tetrahedron and octahedron sheets. Equation (3.5) describes dehydroxylation, which is secondary. Equation (3.6) describes $H_2O$ radiolysis. Equations (3.7)–(3.10) describe OH introduction, where Equations (3.7) and (3.8) describe Si−OH bond regeneration, probably illustrating hydrophilicity recovery by extra irradiation; Equations (3.9) and (3.10) describe OH introduction in the octahedron sheet, probably illustrating lattice plane shrink. Although equation (3.5) is secondary, this reaction probably illustrates lattice plane recovery at 1000 kGy for OH destruction.

# 4. Conclusion

Structure transformation of muscovite single crystal under γ-ray irradiation at 0−1000 kGy was studied with regard to chemical/crystalline structure and surface properties. After irradiation, the chemical/crystalline structure varied obviously, the $H_2O$ amount declined, and surface hydrophilicity declined at low doses. Nevertheless, surface morphology varied slightly. The main results show that the Si−OH bond was removed at low doses, then the Al−OH bond was regenerated at high doses. Simultaneously, crystal shrink occurred at doses lower than 500 kGy. For the 500 kGy-irradiated sample, the (004) lattice plane shrunk by nearly 0.5%. Additionally, CA increased from 20° to 40° with dose increases up to 100 kGy and then became constant with the dose continuously increasing. From the chemical/crystalline structure and surface hydrophilicity analysis, it can be seen that structure transformation is obvious at low doses, while at high doses it is uncertain. That means muscovite is sensitive to low-dose irradiation but not to high-dose (e.g. 500 kGy). Additionally, the main mechanisms were speculated. It seems the main mechanisms involve framework break and $H_2O$ radiolysis. These procedures occurred synchronously and framework break seems dominant at low doses while at high doses $H_2O$ radiolysis is dominant. For framework break, the Si−OH bond was removed and surface hydrophilicity declined. For $H_2O$ radiolysis, extra OH was introduced, $H_2O$ amount declined and lattice plane shrink occurred. The finding-framework break seems dominant at low doses while at high doses $H_2O$ radiolysis is meaningful for realizing the procedure for defect formation within the matrix of clay used for HLRW disposal in practice.

Some drawbacks of this work are that partial intermediate products were not certified and the main mechanisms were speculated, so further research is needed. However, through this work, we could conclude that muscovite is insensitive to high-dose irradiation. This finding is meaningful for detector development especially for detecting high-dose irradiation. Simultaneously, the mechanism is beneficial for realizing the procedure for defect formation within the matrix of clay used for disposal of HLRW in practice, which is of great significance.

Data accessibility. Muscovite sample was brought from University of Cambridge, U.K. Its accurate composition has been analysed and cited as MUS2 in the literature [58,59]. All experimental data were reported in figures 1−9. There are no additional data.

Authors' contributions. H.L.W. carried out the experiment and drafted the paper. Y.P.S., J.C. and X.W. participated in the discussion. M.Z. supported material and also participated in the discussion. All authors discussed the results and commented on the manuscript.

Competing interests. There are no conflicts to declare.

Funding. This work was financially supported by China Postdoctoral Science Foundation (grant nos. 2018M633634XB).

Acknowledgements. We are grateful to the Co-60 γ ray source operating team (Institute of Nuclear Physics and Chemistry, China Academy of Engineering Physics, Mianyang, China) for helping in γ-ray irradiation. Simultaneously, we are grateful to Mr Yang (Chenguang Yang, Shanghai Institute of Applied Physics, Chinese Academy of Sciences, Shanghai, China) for helping in FT-IR, XRD and SEM characterization.

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
