## [Reviewer comments · Royal Society Open Science]

Review History

RSOS-190594.R0 (Original submission)

Review form: Reviewer 1

Is the manuscript scientifically sound in its present form?

No

Are the interpretations and conclusions justified by the results?

No

Is the language acceptable?

No

Is it clear how to access all supporting data?

No

Do you have any ethical concerns with this paper?

No

Have you any concerns about statistical analyses in this paper?

No

Recommendation?

Major revision is needed (please make suggestions in comments)

Comments to the Author(s)

Review comments of the paper entitled "Intensive study on structure transformation within muscovite single crystal under high dose γ ray irradiation and mechanism speculation"

In this work, the authors studied the mechanism of structural transformation with gamma-irradiation on muscovite single crystal. Their aim was to explore the sensibility of muscovite at high dose irradiation and to understand mechanism for defects formation within matrix of clay. However, Minor revision is required before publication.

Abstract:

- Abstract is described very well but, it will be better if it can rewrite in short and more effective words

Introduction

- It is recommended to discuss about the conventional materials used for the detection of high dose radiation.
- It will be better if the authors can refer and discuss the works clay composites of Andrzej Nowicki et al. Thierry Allard et al. R Celis e tal. Karger Kocis et al. Runcy et.al, Poornima Vijayan et.al, Thomaskutty et.al and S.Anilkumar et.al, Hanna J Maria et al. Ranimol Stephen et al. and R. F. Kamaliev et al. and Claudio Colombo et al. in nanostructures and nano-objects)
- Altogether the introduction part is explained well by the authors

Materials and methods

- The authors satisfactorily explained the materials preparation and characterisation techniques.

Result and discussion

- Page 10 fig 1. Please name the insight Raman spectra graph shown.
- Page 14 line 41, the absence of Si-OH peak in the FTIR transmission spectra is explained by the difference in methods and sample structure, it would be nice if it can support with literatures.
- Page 15. Line 32, "partial Si-OH bonds can be generated coupled with Al-OH bonds generation" is there any other characterisation possible to support the regeneration of Si-OH bond at higher gamma dose.
- It is appreciated that the authors concluded the FTIR and Raman analysis well
- The XRD analysis support the dehydroxylation effect of the FTIR result.
- The TGA analysis shows volatilisation of H₂O in the material. At higher dose 1000kGy what is the reason for increasing in volatilisation of H₂O compare to 500kGy
- The contact angle analysis shows decrease in hydrophilicity for low dose irradiation. But it is almost same for 500 and 1000kGy, why? It is contradicting the above TGA studies
- It is appreciated that the mechanism was explained very well by the authors by using equations and schematic diagrams
- Conclusion is written well by the authors, however it will be nice if it can explain much shorter words.

Altogether, the authors made a good attempt to understand the influence of gamma irradiation on muscovite single crystal. This paper can be accepted with minor correction.

Review form: Reviewer 2

Is the manuscript scientifically sound in its present form?

Yes

Are the interpretations and conclusions justified by the results?

Yes

Is the language acceptable?

No

Is it clear how to access all supporting data?

Not Applicable

Do you have any ethical concerns with this paper?

No

Have you any concerns about statistical analyses in this paper?

No

Recommendation?

Accept with minor revision (please list in comments)

Comments to the Author(s)

This is an interesting paper and the data provided will be useful. However, the language is poor in any places and needs to be refined/improved.

The following mandatory revisions are suggested

1. In the discussion on contact angle it is important that the authors provide actual surface roughness values. AFM images of the surfaces will add to the quality of interpretation.
2. The SEM images in figure 7 make no sense. Nothing is visible. Authors should provide better quality images.
3. In the discussion on interlayer spacing, d , the change is attributed to lattice shrinkage. What about strain? It also appears that the relative intensities of the various peaks are changing with irradiation. Can the authors comment on this?

With these minor revisions the paper can be accepted for publication.

Decision letter (RSOS-190594.R0)

10-May-2019

Dear Dr Wang:

Title: Intensive study on structure transformation within muscovite single crystal under high dose γ ray irradiation and mechanism speculation

Manuscript ID: RSOS-190594

Thank you for submitting the above manuscript to Royal Society Open Science. On behalf of the Editors and the Royal Society of Chemistry, I am pleased to inform you that your manuscript will

be accepted for publication in Royal Society Open Science subject to minor revision in accordance with the referee suggestions. Please find the reviewers' comments at the end of this email.

The reviewers and handling editors have recommended publication, but also suggest some minor revisions to your manuscript. Therefore, I invite you to respond to the comments and revise your manuscript.

Please also include the following statements alongside the other end statements. As we cannot publish your manuscript without these end statements included, if you feel that a given heading is not relevant to your paper, please nevertheless include the heading and explicitly state that it is not relevant to your work. We have included a screenshot example of the end statements for reference.

- Ethics statement

Please clarify whether you received ethical approval from a local ethics committee to carry out your study. If so please include details of this, including the name of the committee that gave consent in a Research Ethics section after your main text. Please also clarify whether you received informed consent for the participants to participate in the study and state this in your Research Ethics section.

OR

Please clarify whether you obtained the necessary licences and approvals from your institutional animal ethics committee before conducting your research. Please provide details of these licences and approvals in an Animal Ethics section after your main text.

OR

Please clarify whether you obtained the appropriate permissions and licences to conduct the fieldwork detailed in your study. Please provide details of these in your methods section.

- Funding statement

Please include a funding section after your main text which lists the source of funding for each author.

Because the schedule for publication is very tight, it is a condition of publication that you submit the revised version of your manuscript before 19-May-2019. Please note that the revision deadline will expire at 00.00am on this date. If you do not think you will be able to meet this date please let me know immediately.

- 1) A text file of the manuscript (tex, txt, rtf, docx or doc), references, tables (including captions) and figure captions. Do not upload a PDF as your "Main Document".

- 2) A separate electronic file of each figure (EPS or print-quality PDF preferred (either format should be produced directly from original creation package), or original software format)
- 3) Included a 100 word media summary of your paper when requested at submission. Please ensure you have entered correct contact details (email, institution and telephone) in your user account
- 4) Included the raw data to support the claims made in your paper. You can either include your data as electronic supplementary material or upload to a repository and include the relevant doi within your manuscript
- 5) All supplementary materials accompanying an accepted article will be treated as in their final form. Note that the Royal Society will neither edit nor typeset supplementary material and it will be hosted as provided. Please ensure that the supplementary material includes the paper details where possible (authors, article title, journal name).

Best wishes,
 Dr Laura Smith
 Publishing Editor, Journals

RSC Associate Editor:
 Comments to the Author:
 (There are no comments.)

RSC Subject Editor:
 Comments to the Author:
 (There are no comments.)

Reviewer comments to Author:

Reviewer: 1

Comments to the Author(s)

Review comments of the paper entitled "Intensive study on structure transformation within muscovite single crystal under high dose γ ray irradiation and mechanism speculation"

In this work, the authors studied the mechanism of structural transformation with gamma-irradiation on muscovite single crystal. Their aim was to explore the sensibility of muscovite at high dose irradiation and to understand mechanism for defects formation within matrix of clay. However, Minor revision is required before publication.

Abstract:

- Abstract is described very well but, it will be better if it can rewrite in short and more effective words

Introduction

- It is recommended to discuss about the conventional materials used for the detection of high dose radiation.
- It will be better if the authors can refer and discuss the works clay composites of Andrzej Nowicki et al. Thierry Allard et al. R Celis e tal. Karger Kocis et al. Runcy et.al, Poornima Vijayan et.al, Thomaskutty et.al and S.Anilkumar et.al, Hanna J Maria et al. Ranimol Stephen et al. and R. F. Kamaliev et al. and Claudio Colombo et al. in nanostructures and nano-objects)
- Altogether the introduction part is explained well by the authors

Materials and methods

- The authors satisfactorily explained the materials preparation and characterisation techniques.

Result and discussion

- Page 10 fig 1. Please name the insight Raman spectra graph shown.
- Page 14 line 41, the absence of Si-OH peak in the FTIR transmission spectra is explained by the difference in methods and sample structure, it would be nice if it can support with literatures.
- Page 15. Line 32, "partial Si-OH bonds can be generated coupled with Al-OH bonds generation" is there any other characterisation possible to support the regeneration of Si-OH bond at higher gamma dose.
- It is appreciated that the authors concluded the FTIR and Raman analysis well
- The XRD analysis support the dehydroxylation effect of the FTIR result.
- The TGA analysis shows volatilisation of H₂O in the material. At higher dose 1000kGy what is the reason for increasing in volatilisation of H₂O compare to 500kGy
- The contact angle analysis shows decrease in hydrophilicity for low dose irradiation. But it is almost same for 500 and 1000kGy, why? It is contradicting the above TGA studies
- It is appreciated that the mechanism was explained very well by the authors by using equations and schematic diagrams
- Conclusion is written well by the authors, however it will be nice if it can explain much shorter words.

Altogether, the authors made a good attempt to understand the influence of gamma irradiation on muscovite single crystal. This paper can be accepted with minor correction.

Reviewer: 2

Comments to the Author(s)

This is an interesting paper and the data provided will be useful. However, the language is poor in any places and needs to be refined/improved.

The following mandatory revisions are suggested

1. In the discussion on contact angle it is important that the authors provide actual surface roughness values. AFM images of the surfaces will add to the quality of interpretation.
 2. The SEM images in figure 7 make no sense. Nothing is visible. Authors should provide better quality images.
 3. In the discussion on interlayer spacing, d , the change is attributed to lattice shrinkage. What about strain? It also appears that the relative intensities of the various peaks are changing with irradiation. Can the authors comment on this?
- With these minor revisions the paper can be accepted for publication.

Author's Response to Decision Letter for (RSOS-190594.R0)

See Appendix A.

RSOS-190594.R1 (Revision)

Review form: Reviewer 1

Is the manuscript scientifically sound in its present form?

Yes

Are the interpretations and conclusions justified by the results?

Yes

Is the language acceptable?

Yes

Is it clear how to access all supporting data?

Yes

Do you have any ethical concerns with this paper?

No

Have you any concerns about statistical analyses in this paper?

No

Recommendation?

Accept as is

Comments to the Author(s)

1. Authors satisfactorily rewritten the abstract
2. The discussion on the conventional material is included in the introduction part
3. The literatures suggested were added satisfactorily
4. The comment on FTIR is well explained by the authors with schematic diagrams
5. Reasonable explanation is given for the limitations in characterization on the regeneration Si-OH

6. Increase in volatilization of H₂O on higher dose is acceptable from the explanations
7. Supporting information's from the FTIR for the comment 12 is reasonable
Authors have addressed all the comments and give detailed explanations. This paper is acceptable for the publication. However, it is recommended to check the grammatical mistakes once again.

Decision letter (RSOS-190594.R1)

12-Jun-2019

Dear Dr Wang:

Title: Intensive study on structure transformation of muscovite single crystal under high dose γ ray irradiation and mechanism speculation
Manuscript ID: RSOS-190594.R1

It is a pleasure to accept your manuscript in its current form for publication in Royal Society Open Science. The chemistry content of Royal Society Open Science is published in collaboration with the Royal Society of Chemistry.

RSC Associate Editor:
Comments to the Author:
(There are no comments.)

RSC Subject Editor:
Comments to the Author:
(There are no comments.)

Reviewer(s)' Comments to Author:

Reviewer: 1

Comments to the Author(s)

1. Authors satisfactorily rewritten the abstract
 2. The discussion on the conventional material is included in the introduction part
 3. The literatures suggested were added satisfactorily
 4. The comment on FTIR is well explained by the authors with schematic diagrams
 5. Reasonable explanation is given for the limitations in characterization on the regeneration Si-OH
 6. Increase in volatilization of H₂O on higher dose is acceptable from the explanations
 7. Supporting information's from the FTIR for the comment 12 is reasonable
- Authors have addressed all the comments and give detailed explanations. This paper is acceptable for the publication. However, it is recommended to check the grammatical mistakes once again.

Appendix A

May 18, 2019

Dear Reviewers,

We would like to express our gratitude to you for your critical reading and reviewing of our manuscript. We very appreciate your kind help and the comments are valuable for us to make revision and constructive to improve our work. We have revised our manuscript carefully according to the comments. Simultaneously, other errors were also corrected to make the manuscript more accurate and informative. The correction details are mainly listed in next section and indicated in blue color in manuscript. **All answers to reviewers are in italic in follows.**

Thanks again!

Yours sincerely,

Honglong Wang, Dr.

Ming Zhang, Dr, Prof.

Reviewer comments to Author:

Reviewer: 1

Comments to the Author(s)

Review comments of the paper entitled “Intensive study on structure transformation within muscovite single crystal under high dose γ ray irradiation and mechanism speculation”. In this work, the authors studied the mechanism of structural transformation with gamma-irradiation on muscovite single crystal. Their aim was to explore the sensibility of muscovite at high dose irradiation and to understand mechanism for defects formation within matrix of clay. However, Minor revision is required before publication.

Abstract:

(1) Abstract is described very well but, it will be better if it can rewrite in short and more effective words

Thanks very much for your kind reading, your suggestion is constructive. This time,

we revise the abstract carefully, partial sentences were deleted, and partial sentences were rewritten, trying to make the abstract more succinct and informative. Additionally, other parts were also revised to make the manuscript succinct.

Introduction

(2) It is recommended to discuss about the conventional materials used for the detection of high dose radiation.

Thanks very much for your suggestion, your suggestion is beneficial for realizing research in this field, which is meaningful. After reading literature, a paragraph was added in this revision in page 4-5 as “.....Nowadays, numerous materials have been evaluated such as polymer, semiconductor (silicon), glass, calcium fluoride (CaF₂), etc [11]. However, they are probably improper for shortcomings. For instance, polymer is easy to degrade and be heated by irradiation [12-14], semiconductor is easy to conduct electricity [12], the composition of glass is complex, CaF₂ is easy to generate toxic gas under irradiation [12]. In this case, designing material which is suitable for detecting high dose irradiation is still challenging. Except for low cost and stability, muscovite has partial advantages such as nice electric insulation, heat isolation [15] and transparency. These advantages are beneficial for storing accumulated effects and observing ion track especially for ion irradiation [5, 16]. It may have a potential application in high dose irradiation detection [17]. Thus, a clear cognition on its sensibility at high dose irradiation is meaningful”. We hope these descriptions could make the manuscript more informative.

(3) It will be better if the authors can refer and discuss the works clay composites of Andrzej Nowicki et al. Thierry Allard et al. R Celis e tal. Karger Kocis et al. Runcy et.al, Poornima Vijayan et.al, Thomaskutty et.al and S.Anilkumar et.al, Hanna J Maria et al. Ranimol Stephen et al. and R. F. Kamaliev et al. and Claudio Colombo et al. in nanostructures and nano-objects)

Thanks very much for your kind reading, your suggestion is constructive. You have read numerous literatures, all their works are on clay structure or applications. We

are very appreciated for your hard work. After realizing their works, we have a more clear cognition on clay structure and its application in engineering material manufacture, environmental science, medical science, etc. A clear description on clay application would make the manuscript more plentiful. In this case, several sentences were added in page 5 as “.....*In reality, for low cost, nice fire resistance and nontoxic, clay is widely used in material, environmental and medical science as engineering material manufacture [36-41], sewage treatment or environmental remediation [42-44], drug delivery [45], etc.*”.

(4) Altogether the introduction part is explained well by the authors

Thanks very much for your kind reading, hard work and positive comment.

Materials and methods

(5) The authors satisfactorily explained the materials preparation and characterisation techniques.

Thanks very much for your kind reading, hard work and positive comment.

Result and discussion

(6) Page 10 fig 1. Please name the insight Raman spectra graph shown.

*Thanks very much for your kind reading, your suggestion is constructive. This time, the inside graph was named and described carefully in page 10 as “.....*Seeing spectra in macro, it might imply great variation in 1000 kGy-irradiated sample for weak intensity. However, that is incorrect as shown inside graph. The inside graph (1000 kGy-irradiated sample) is generally similar to curves of other samples.*”.*

(7) Page 14 line 41, the absence of Si-OH peak in the FTIR transmission spectra is explained by the difference in methods and sample structure, it would be nice if it can support with literatures.

This is a very good suggestion. In reality, we have tried our best to search some literatures to support this viewpoint. However, little literatures were found. This is

due to some difficulties. Normally, for FT-IR transmission-model experiment, the propagation direction of photon is vertical to sample surface, the angle of projection is 90° . For FT-ATR experiment, the propagation direction of photon inclines to sample surface normally with an angle of projection as 45° . Main difference within these two methods is that the angle of projection is different. If we can alter the angle of projection, the spectrum might vary obviously and certify this viewpoint. However, main angle of projection for commercial FT-IR spectrum instrument is constant as the sample stage is fixed. The FT-IR spectrum instrument with rotatable sample stage is not common. In this case, literature on peak intensity vs the angle of projection is rare. Nevertheless, the orientation of OH can be realized. According to literature, the C*-OH angle can be considered as less than 30° ($0\sim 30^\circ$). This can be seen in manuscript on page 15 as “.....For muscovite, its OH vector in z direction is strong (the included angle between z-direction and OH is less than 30°) [71], and Si-OH vibration is weaker compared to Al-OH.....”. In this case, it can be explained as follows. To have a clear understanding, two figures will be shown.

Fig. 1 shows a scheme for interaction with OH vibration and projected photon in FT-ATR experiment. Line a represents the propagation direction of projected photon, which inclines to surface with an angle as 45° . The surface could be described as X-Y plane, z-direction is vertical to this plane. According to literature, the OH vector is close to z-direction with an angle less than 30° . Assuming the angle is 30° , which could be described as line b. In this case, the vector \overrightarrow{OH} could represent OH orientation in sample. This vector can be described as two vectors overlay ($\overrightarrow{OH} = \overrightarrow{OM} + \overrightarrow{MH}$). As the propagation direction of projected photon is in line a, the vibration orientation can be described in line a' plane. In this case, only the vector for OH vibration in line a' plane can interact with photon, leading spectrum absorption. On aforementioned assumption, the included angle for OH vector and line a' plane can be described as 15° ($45^\circ - 30^\circ$). Assuming the intensity for OH vibration as I_0 , the intensity for OH vibration in \overrightarrow{OM} orientation as I_1 , thus, $I_1 = I_0 \cos 15^\circ = 0.97I_0$, That means the intensity for OH vibration which can interact

with projected photon is very strong close to 100%. In other words, the signal in FT-ATR spectrum would be intense. The OH absorption spectrum would be obvious.

Fig. 1 A scheme for interaction with OH vibration and projected photon in FT-ATR experiment.

Fig. 2 A scheme for interaction with OH vibration and projected photon in FT-IR transmission experiment.

Fig. 2 shows a scheme for interaction with OH vibration and projected photon in FT-IR transmission experiment. Line a represents X-Y plane, which could represent sample surface. The propagation direction of projected photon is in z-direction, which is vertical to sample surface (line a). As the included angle for OH vector with z-direction is close to 30°, thus, line b could describe the orientation for OH vector. That means vector \overrightarrow{OH} could represent orientation of OH in sample. This vector can be described as two vectors overlay ($\overrightarrow{OH} = \overrightarrow{OM} + \overrightarrow{MH}$). As the propagation direction of projected photon is in z-direction, the vibration orientation for photon can be

described in line a' plane (X-Y plane). In this case, only the vector for OH vibration in line a' plane or normal to this plane can interact with photon, leading spectrum absorption. On aforementioned assumption, the included angle for OH vector and z-direction can be described as 30° . Assuming the intensity for OH vibration as I_0 , the intensity of OH vibration in \overline{MH} orientation as I_1 , thus, $I_1 = I_0 \sin 30^\circ = 0.5I_0$. That means the intensity for OH vibration which can interact with projected photon is 50 percent whole intensity.

Seeing aforementioned analysis, for muscovite in FT-ATR experiment, the intensity for OH interacting with projected photon is close to whole intensity. Nevertheless, in FT-IR transmission-model experiment, the intensity for OH interacting with projected photon is 50 percent whole intensity. That means for constant sample the OH vibration absorbance in FT-ATR spectrum would be more intense compared to FT-IR transmission spectrum. In other words, the OH signal in FT-IR transmission-model experiment would be weaker compared to that in FT-ATR spectrum for muscovite.

Normally, seen from formula- $KAl_2(AlSi_3O_{10})(OH)_2$ and structure, the OH should be Al-OH, none Si-OH bond should be observed. Nevertheless, for partial defects existence, Si-OH bond can be observed in silicate such as SiO_2 , in this case, Si-OH vibration can be observed. Normally, the amount for Si-OH bond is tiny, the majority for OH is displayed as M-OH bond. In this work, the majority is Al-OH bond. For its tiny amount, probably less than 1%, its orientation wasn't reported. In most cases, it was reported as OH orientation didn't showing distinguish. As the signal for Al-OH vibration in FT-IR transmission-model experiment is weaker compared to that in FT-ATR experiment, assuming the orientation for OH linked to Si atom is close to that linked to Al. In this case, the signal for Si-OH vibration in FT-IR transmission-model experiment would be very weak. Additionally, its orientation might in z-direction. In this case, the signal for Si-OH vibration might disappear in FT-IR transmission spectrum. These explanations could probably explain why Si-OH band wasn't observed in FT-IR transmission-model experiment. Generally, a clear cognition on orientation for OH linked to Si atom is difficult because of low amount.

Simultaneously, for low amount, its signal can be weak. A clear exploration is meaningful and the suggestion is constructive.

(8) Page 15. Line 32, “partial Si-OH bonds can be generated coupled with Al-OH bonds generation” is there any other characterisation possible to support the regeneration of Si-OH bond at higher gamma dose.

Thanks very much for your kind reading, your question is meaningful and very scientific. A clear answer to this question is challenging, which needs more research. Huge difficulty exists in solving this problem.

Firstly, the linkage break between tetrahedron and octahedron sheet is the break of Al-O-Si or Al-O-Al bond in nature. Under Al-O-Si bond break, Al-O• and Si• radicals or Al• and Si-O• radicals would be generated. If the Al-O• or Al• radicals form Al-OH bond, to make system stable, the Si• or Si-O• radicals might transform to Si-OH bond finally. This is the meaning of- “partial Si-OH bonds can be generated coupled with Al-OH bonds generation”. These reactions are in micro and the broken part is tiny for low linear energy transfer effect for γ ray irradiation. Finally, the extra introduced OH amount would be tiny. Simultaneously, Al-O-Al bond might be destroyed forming Al-OH bond. In reality, during irradiation process, we cannot assure whether Al-O-Si or Al-O-Al bond occurred break but could assure the probability for Al-OH formation deriving from linkage break is higher compared to Si-OH bond formation. That means the amount for Si-OH introduced would be tiny. For tiny amount, a clear cognition on orientation for extra introduced Al-OH bond is difficult. In this case, a clear cognition on Si-OH bond would be more difficult.

Secondly, the reaction for γ ray with framework atom is random. For vacancy existence, the orientation for extra introduced Si-OH bond may be various as the position for H• or OH• radicals cannot be assured. Thirdly, the tetrahedron sheet can be destroyed and introduced Si-OH bond. The orientation for these Si-OH bonds might be random. In this case, it is difficult to assure the Si-OH bond has constant orientation. In this case, it is very difficult to distinguish whether the Si-OH bond is originally existed or derived from linkage break or tetrahedron destruction from

orientation difference. A clear distinguish on this variation in molecule level is difficult. Maybe this procedure can be observed in future in situ, which needs huge advance in technology. If this procedure can be observed in situ, the viewpoint can be certified more accurately.

Normally, the wave number can be constant as it seems independent on orientation. In this case, FT-IR spectrum can describe Si-OH bond variation. Although, we cannot distinguish the difference, we can have a qualitative cognition on structure variation.

Except for FT-IR, NMR can be used to describe micro condition difference for Si element. However, Si-OH bond originally exists. Even after irradiation, none extra signal probably can be observed as Si-OH bond originally existed in tetrahedron sheet or deriving from linkage break may show similar split. It is difficult to assure which signal derives from linkage break.

Generally, having a clear observation in micro and observing this procedure in situ is difficult. However, this question is beneficial for micro procedure cognition, which is important. Solving this problem needs technology advance and efforts. We think this question can be answered in future.

(9) It is appreciated that the authors concluded the FTIR and Raman analysis well.

Thanks very much for your kind reading, hard work and positive comment.

(10) The XRD analysis support the dehydroxylation effect of the FTIR result.

Thanks very much for your kind reading, hard work and positive comment.

(11) The TGA analysis shows volatilisation of H₂O in the material. At higher dose 1000 kGy what is the reason for increasing in volatilisation of H₂O compare to 500 kGy.

Thanks very much for your kind reading, this is a good question. In reality, seen from TGA curves, the residue mass for 1000 kGy-irradiated sample is lesser compared to 500 kGy-irradiated sample. That means the volatilization of H₂O in 1000 kGy-irradiated sample is higher than 500 kGy-irradiated sample. That means H₂O

amount in 1000 kGy-irradiated sample is higher than that in 500 kGy-irradiated sample. This can be explained as follows. Normally, the procedure for H₂O amount variation is complex. Upon irradiation, two procedures occurred synchronously. One is chemical bond break such as the break of Al-OH bond shown as dehydroxylation, raising H₂O amount. The other is extra group's introduction related to H₂O radiolysis, reducing H₂O content. They had opposite effects on H₂O amount variation. At low dose, framework break is dominant and H₂O radiolysis can be promoted by dose increase. In this case, H₂O amount could reduce obviously. At high dose, extra irradiation probably results in more break of Al-OH, and H₂O can be regenerated. In this case, H₂O amount could be close to that in pristine sample. In this case, 500 kGy-irradiated sample could have lower H₂O amount compared to 1000 kGy-irradiated sample. Generally, that is probably because 500 kGy-irradiated sample has more intense H₂O radiolysis level compared to 1000 kGy-irradiated. In this case, H₂O amount in 500 kGy-irradiated sample could be lower than that in 1000 kG-irradiated sample.

This question has been discussed carefully in manuscript in pages 23-24 as "It seems H₂O amount in 500 kGy-irradiated sample is lower *than* that in 1000 kGy-irradiated sample and H₂O amount in 1000 kGy-irradiated sample is close to that in pristine sample. These results can support FT-IR analysis commendably as 500 kGy-irradiated has the most intense signal for OH vibration (**Fig. 3**). Simultaneously, if we assume lattice plane shrink is completely ascribed to extra OH introduction, in this case, 500 kGy-irradiated sample has the most intense shrink in lattice plane. In other words, it has the largest amount of OH introduction. The extra *introduced* OH *dues* to H₂O radiolysis. Thus, 500 kGy-irradiated sample should have lower H₂O amount compared to pristine sample. TGA analysis certified this assumption. That means TGA results supported XRD analysis indirectly. The reason for 500 kGy-irradiated sample has lower H₂O amount compared to pristine *and* 1000 kGy-irradiated samples can be explained as follows. Upon irradiation, two procedures occurred synchronously. One is chemical bond break such as the *break* of Al-OH bond shown as dehydroxylation. The other is extra group's introduction related

to H₂O radiolysis. The former procedure *would* raise H₂O amount, the latter *would* reduce H₂O amount. Finally, mechanism for H₂O amount variation is complex. *At low dose*, framework break is dominant and H₂O radiolysis can be promoted by dose increase. *At high dose*, extra irradiation probably results in more break of Al-OH, and H₂O can be regenerated. In this case, 500 kGy-irradiated sample might have lower H₂O amount compared to pristine and 1000 kGy-irradiated samples and H₂O amount in 1000 kG-irradiated sample can be close to that in pristine sample. This phenomenon is normally observed as radiation effect isn't linearly with absorbed dose such as radiation crosslinking of ultra-high molecule weight polyethylene sheet [65]. After a threshold, the degree of cross-linking would be constant or smaller.”

(12) The contact angle analysis shows decrease in hydrophilicity for low dose irradiation. But it is almost same for 500 and 1000kGy, why? It is contradicting the above TGA studies

Thanks very much for your kind reading, this is a very good question. Seen from contact angle analysis, the hydrophilicity declined at low dose while at high dose such as at 500 and 1000 kGy is similar. Seen from TGA analysis, 500 kGy-irradiated sample had lower H₂O amount compared to 1000 kGy-irradiated sample. In other words, it has more intense H₂O radiolysis level. Finally, it had more extra OH introduction. This conclusion can be certified by FT-IR spectrum as 500 kGy-irradiated sample had more intense OH signal compared to 1000 kGy-irradiated sample. Generally, seen from FT-IR and TGA analysis, 500 kGy-irradiated sample had more extra OH introduction compared to 1000 kGy-irradiated sample, while the hydrophilicity for these two samples were similar. It seems confusing. This can be explained as follows. Firstly, the extra introduced OH might exist in octahedron sheet as seen from FT-IR spectrum. In reality, the effect for this introduction on elevating surface hydrophilicity is weak as surface hydrophilicity mainly relies on surface structure (e.g. tetrahedron sheet, interlayer ions). Secondly, extra OH can be introduced on tetrahedron sheet, which could promote surface hydrophilicity. Thirdly, the surface destruction cannot be avoided. These procedures might even have

opposite effect on surface hydrophilicity variation. Finally, the procedure for hydrophilicity is complex. In this case, although 500 kGy-irradiated sample had more intense H₂O radiolysis level compared to 1000 kGy-irradiated sample, their surface hydrophilicity can be similar.

In reality, surface hydrophilicity variation has a complex mechanism involving OH amount variation, tetrahedron sheet destruction, roughness variation, etc. Finally, the variation maybe in-linearly with OH amount variation or absorbed dose. Additionally, for hydrophic material, after a threshold, the hydrophilicity wouldn't increase with OH amount increase. For instance, radiation grafting polyacrylic acid to increase surface hydrophilicity of a polymer or CNTs, after a threshold, the CA wouldn't decrease or even increase with increase of degree of grafting. This phenomenon could also explain why the surface hydrophilicity for these samples was similar. These explanations could probably explain the conflict.

(13) It is appreciated that the mechanism was explained very well by the authors by using equations and schematic diagrams.

Thanks very much for your kind reading, hard work and positive comment.

(14) Conclusion is written well by the authors, however it will be nice if it can explain much shorter words.

Thanks very much for your kind reading, hard work and positive comment. We have revised it carefully to make it more succinct and informative.

(15) Altogether, the authors made a good attempt to understand the influence of gamma irradiation on muscovite single crystal. This paper can be accepted with minor correction.

Thanks very much for your kind reading, hard work and positive comment. We have revised it completely. All suggestions are constructive, benefiting improving the manuscript and research. Except for these suggestions, other parts or errors were also corrected. Additionally, in this response, partial viewpoint should be certified by

literature, we have tried this strategy. However, this introduction would make the description confusing as partial revised part of main manuscript is displayed in this file. Quoting can be seen. If we extra cite partial literature, the file would be difficult to read. Generally, the explanation is plentiful and assured. Via considering these questions, we have a deeper realization on our work, which improved our research sufficiently. We would show our appreciation to you for your hard work and help once again!

Reviewer: 2

Comments to the Author(s)

(1) This is an interesting paper and the data provided will be useful. However, the language is poor in any places and needs to be refined/improved.

Thanks very much for your kind reading, hard work and pertinent comment. The language is our shortcoming as english isn't our native language. Although we have studied english for several years, trying to use english to communicate with other people or write articles with completely correct grammar or native words is still difficult. Normally, improving the english communication level is a hard work, which needs long time training and accumulation. This is our shortcoming, we will try our best to improve english level, trying to make the communication more relevant. This time, we revise our manuscript carefully, main part has been rewritten, trying to make the manuscript easily understandable and accurate. We will try our best to improve english communication skill in future. Thanks very much for your suggestion.

(2) In the discussion on contact angle it is important that the authors provide actual surface roughness values. AFM images of the surfaces will add the quality interpretation.

Thanks very much for your kind reading and suggestion. The suggestion is constructive and meaningful to realize variation in micro. In this case, AFM experiments were performed and roughness values were calculated and discussed and

shown in pages 9 and 29-33 in this revision. For revision, **Figs. 8 and 9** were added. Main revision can be displayed as follows.

Page 9 shows experimental method as “**Atomic force microscopic (AFM)**. AFM experiments were performed on a NTEGRA Prima instrument (NT-MDT Co.). Prior to measurement, a thin layer of double faced adhesive tape was fixed on glass slide, then, film was fixed on tape. Tapping mode of scanning was adopted and datum was analyzed by Nova-px software.”

Pages 29-33 discussed results as “To observe morphology more clearly, AFM experiments were performed. **Fig. 8** shows AFM images for surface of pristine and irradiated samples. Generally, variation in z-direction is tiny as in nanometer scale, implying smooth. To have a clear realization on difference in z-direction, an area is chosen randomly as shown in green color in image (**Fig. 8**). Then, height in z-direction after cropping vs distance along green line is displayed and shown in **Fig. 9**. Seeing **Fig. 9a**, variation range after cropping for pristine sample is close to ± 2 nm except for partial section. Seeing **Figs. 9b-d**, variation ranges after cropping for 100 kGy-irradiated, 200 kGy-irradiated and 500 kGy-irradiated samples can be considered as ± 0.7 , ± 0.2 and ± 0.6 nm, respectively. Seeing **Fig. 9e**, variation range after cropping for 1000 kG-irradiated sample is close to ± 1.5 nm. For pristine sample (**Fig. 9a**), partial section shows large variation range. This is probably due to defect originally existed as sample is natural and in sheet form. In this case, surface might exist bump or depression induced by mining. Additionally, film is peeled randomly. If sample contains defects in internal, partial sections can be broken during peeling. In this case, partial bump or depression can be observed. Except for that region, sample is generally smooth with a variation range close to ± 2 nm in z-direction in chosen part. For 100 kGy-irradiated, 200 kGy-irradiated and 500 kGy-irradiated samples, the variation ranges are close to ± 0.7 , ± 0.2 and ± 0.6 nm. They are very tiny, showing smooth. In reality, the size of T-O-T structure in z-direction is close to 1.2-1.5 nm, chemical bond length for K-O ionic bond or Si-O/Al-O bond is close to 0.2 nm. Seeing value especially for 200 kGy-irradiated

sample, the variation range is close to ± 0.2 nm, which is close to the length of chemical bond as K-O ionic bond or Si-O bond. Normally, the distribution of interlayer ion (e.g. K^+ ion) is random, in this case, the variation in z-direction at atom level may be close to 0.2 nm. In this work, the variation range is close to this scale. That means the surface is very smooth.

Generally, for all samples, the variation range in z-direction after cropping is less than ± 2 nm. The range is tiny. Normally, variation range induced by artificial factor is large, which can be in μm scale or larger. Trying to induce variation in nanometer scale by artificial technology is difficult. In reality, this tiny variation needs special instrument or technology, which should be processed precisely such as by etching of laser. For sample peeled by manual, this tiny variation can be endured. Generally, this tiny variation indicates sample is smooth, which further certified SEM results.

Although sample is smooth, variation range for pristine sample seems larger than other samples. That means it seems rougher than other samples as roughness can be described by variation range in z-direction. The reason can be ascribed to sample preparation or difference. For irradiated samples, they seem smoother compared to pristine sample especially for 200 kGy-irradiated sample. That cannot be ascribed to irradiation as it is difficult to prepare sample with constant surface structure by manual. Simultaneously, we cannot assure the defect in sheet is uniform. Additionally, variation induced by irradiation in z-direction would be very tiny seen from aforementioned explanations (SEM part). Its variation maybe covered by sample preparation or difference. In this case, apparent roughness difference cannot be ascribed to irradiation process. If we consider pristine sample has largest roughness and all samples have similar chemical structure in surface, in this case, irradiated samples should have larger CAs especially for 200 kGy-irradiated sample as this sample seems have the smallest roughness. Nevertheless, obtained CA is contrary to this expectation. They have close CAs (**Fig. 6**). That means great variation existed in chemical structure. In other words, CA variation wasn't aroused by surface morphology difference but to chemical structure variation, which means aforementioned analysis on chemical structure is probably suitable.

Generally, seen from SEM and AFM analysis, it concludes the CA variation was attributed to intrinsic structure *difference* not to artificial factor. In reality, numerous procedures could affect surface hydrophilicity such as framework break, Si-OH removal or regeneration, roughness variation, etc. *They might* show *offset* effects on CA *variation*. Finally, CA didn't vary linearly vs absorbed dose.

Generally, low dose irradiation sufficiently declined hydrophilicity while extra irradiation recovered. Irradiation almost had none effects on surface morphology.”

Fig. 8 AFM images for surface of muscovite under γ ray irradiation at 0-1000 kGy.

Fig. 9 Height in z-direction after cropping vs distance along chosen area for different samples.

(3) The SEM images in figure 7 make no sense. Nothing is visible. Authors should provide better quality images.

Thanks very much for your kind reading, and constructive suggestion. This suggestion is important. In reality, we have tried our best to make the image more clearly. However, partial difficulties exist. This can be illustrated as follows. In reality, we have observed surface morphology by SEM several times and obtained numerous images by an advanced instrument. The instrument has high resolution. However, the images are similar. That means the quality for image wasn't up to instrument or staff skills but to sample intrinsic characteristic. This can be explained as follows. Firstly, the sample is electric insulation and transparent, it is difficult to conduct charge. In this case, numerous charges are deposited showing highlight. This phenomenon is common in SEM characterization. If the material has better electrical conductivity, the image would be dark. Secondary, the sample is smooth and uniform. There is no great difference in composition or size in height. If the surface has different composition in distinct position, the charging effect would be different. In this case, the image would have huge contrast. This phenomenon is common in composite. Simultaneously, if the surface has great variation in size of height, the charging effect would also different. In this case, huge contrast can also be observed. This phenomenon is common in observing cross-section for sample after break under stress or composite. As sample with poor electrical conductivity, transparent, and uniform composition and smooth surface, the contrast is small. In this case, the image is highlight and seems show none great difference. These descriptions could probably explain the obtained image with tiny contrast.

In reality, this phenomenon is meaningful, the obtained image is informative. If there were cracks or grooves or impurities on sample surface, the contrast would be huge. In this case, partial parts would be dim. The highlight image with tiny contrast means none cracks, grooves or impurities (e.g. particles) exist on sample surface, which is expected.

(4) In the discussion on interlayer spacing, d , the change is attributed to lattice

shrinkage. What about strain? It also appears that the relative intensities of the various peaks are changing with irradiation. Can the authors comment on this?

Thanks very much for your kind reading and hard work. This is a very good question. Seen from XRD analysis, the interlayer space d became smaller under irradiation, meaning shrink. Under 500 kGy irradiation, interlayer space d for (004) lattice plane shrunk near 0.5%, which is close to 0.02 Å. In reality, the variation range is strain. In this case, we could consider the strain for (004) lattice plane under 500 kGy irradiation as 0.5%. This can be seen in XRD analysis in page 18 as “…… Generally, under 500 kGy irradiation, (004) lattice plane shrunk near 0.5%, closing to 0.02 Å ($4.985 (\pm 0.002) - 4.962 (\pm 0.002) = 0.023 (\pm 0.002)$ Å). ……”. Except for strain, partial stress might exist. Seeing XRD analysis, after irradiation, diffraction angle became larger compared to pristine sample showing shrink, meaning tension force existed. This is normal as hydrogen bond formation strengthens the interlayer force between tetrahedron and octahedron sheet. Generally, tension stress existed, which is expected. Finally, shrinkage was observed.

Seen from XRD patterns, relative intensities for various peaks varied after irradiation. This is interesting, and can be explained as follows. Normally, for sample irradiated by γ ray, the crash between photon and framework is random. In this case, which bond occurred break and position is uncertain. For instance, we cannot assume the destruction mainly in (002) lattice plane or (004) lattice plane. In reality, (002), (004), (006) and (008) lattice planes are in z -direction, and similar and parallel. The variation may be random because of layered structure. Simultaneously, the variation part is tiny. Additionally, the variation can even be recovered during irradiation process. In this case, the reason for relative intensity for peak variation is difficult to explain. Although, the variation in relative intensity is difficult to explain, the variation in diffraction angle is interesting. For instance, if the diffraction angle becomes larger meaning shrink, if the diffraction becomes smaller meaning expansion. Additionally, if the interval position appears peak meaning serious decomposition or phase transformation. These variations are essential for realizing radiation damage, which is of great significance. Generally, variation trend is similar. In this case, we

could conclude sample occurred shrink after irradiation.

(5) With these minor revisions the paper can be accepted for publication.

Thanks very much for your kind reading, hard work and positive comments. Almost all suggestions were adopted, they are important to improve our research. Simultaneously, partial extra parts were revised and almost the whole manuscript was rewritten. We hope it could be easily understandable and more accurate. In reality, in this file for answer to question, partial literatures should be cited to make the answer more plentifully. We have tried this strategy, however, which makes the description confusing as partial revised main text in this file. Numerous literatures were numbered in that part. In this case, if we cite extra literature, the file would be difficult to read. Generally, the answer is plentiful and assured. Generally, via answering these questions, we have a deeper realization on our shortcoming and our work, which improved our research sufficiently. We would make the research more deeply and systematically in future. We hope we could obtain your guidance and help more, trying to improve our academic level finally. We would show our appreciation to you for your hard work once again!